# IntervalGP-VAE: Learning Unobserved Confounders with Uncertainty for Personalized Causal Effect Estimation

## Abstract

Estimating individual treatment effects (ITEs) in the presence of unobserved confounding remains a central challenge in causal inference. Existing proxy-based methods aim to recover latent confounders from observational proxies, but typically produce only point estimates without uncertainty quantification. This lack of uncertainty modeling leads to incomplete and potentially insufficient information for downstream decision-making, especially when uncertainty is inherent in the data. We propose IntervalGP-VAE, a novel framework that combines variational autoencoders with Gaussian Process (GP) to model both the latent confounders and their associated uncertainty. At the core of our method is an interval-valued GP prior, which enables the model to capture a distribution over plausible latent confounders and treatment responses, rather than relying on potentially unreliable point estimates. This approach accounts for uncertainty arising from noisy and imperfect proxy variables and yields calibrated ITE interval to support more robust causal decisions. We provide theoretical guarantees for identifiability of the latent confounder up to a smooth monotonic transformation under weak assumptions. Experiments on synthetic and semi-synthetic datasets demonstrate that IntervalGP-VAE achieves superior performance in ITE estimation and uncertainty calibration, outperforming existing methods.

## 1 Introduction

Estimating Individual Treatment Effects (ITEs) from observational data is a core challenge, especially under unobserved confounding (Pearl, 2009; Peters et al., 2017). Recent advances have enabled the inference of latent confounders from proxy variables Louizos et al. (2017); Zhang et al. (2021); Wu et al. (2024); Harada & Kashima (2024), yet significant *uncertainty* remains in this inference process. Proxy variables are often noisy and only weakly related to the true confounders; limited data or poor proxy quality further amplifies this uncertainty. In decision support, quantifying the uncertainty in recovered latent structure and treatment effect estimates is crucial. Modeling the latent confounder as an interval-valued variable captures and propagates uncertainty to counterfactual and ITE estimates.

Recovering unobserved confounders and estimating ITEs under uncertainty requires explicit uncertainty quantification throughout the pipeline from latent confounder inference to outcome prediction. The model must also preserve spatial or structural coherence in the latent space, enabling smooth transitions across similar individuals while capturing heterogeneity. These demands call for flexible, nonparametric probabilistic models. Gaussian Processes (GPs), which inherently model smoothness, uncertainty, and spatial correlation, are a natural fit (Rasmussen & Williams, 2006). However, standard GPs assume fully observed inputs and outputs, whereas our setting requires a principled integration of deep latent variable models with uncertainty quantification and spatial coherence under interval constraints in the latent space.

We propose *IntervalGP-VAE*, a novel framework that integrates interval-valued Gaussian Processes with variational autoencoders (VAEs) to recover unobserved confounders with uncertainty for personalized causal effect estimation. The model employs a VAE encoder to infer a structured latent representation from noisy proxy measurements, enabling individualized confounder modeling under

uncertainty. An interval-valued GP prior is imposed over this latent representation to model confounders as interval-valued function of the proxies. Outcome prediction is then performed via an interval-valued GP regressor, which maps the inferred confounder intervals to calibrated outcome bounds. This enables principled uncertainty propagation and supports smooth, individualized ITE estimation with calibrated confidence intervals, accounting for both latent uncertainty and proxy noise. To our knowledge, IntervalGP-VAE is the first method that combines proxy-based latent variable modeling with interval-valued GPs. Although GPs are increasingly used, few methods handle interval-valued data, and none incorporate GP priors for confounder recovery from proxies. Existing approaches such as CEVAE (Louizos et al., 2017) typically rely on standard VAE priors (e.g., isotropic Gaussians) and lack structured uncertainty propagation grounded in structured priors.

Our main contributions are summarized as follows:

- **Theory:** We present a theoretical analysis of the identifiability conditions under which latent confounders can be recovered from proxies with uncertainty, offering formal guarantees for the proposed method.
- **Methodology:** We propose IntervalGP-VAE, a novel framework that combines VAEs with interval-valued GPs to disentangle latent confounders and measurement noise from noisy proxies and quantify predictive uncertainty. The model integrates an interval-valued GP prior over the latent space and a GP-based interval likelihood head to enable smooth and uncertainty-aware estimation of counterfactual outcomes and treatment effects.
- **Empirics:** We evaluate on 24 synthetic settings constructed to satisfy the identification conditions, and on the semi-synthetic IHDP benchmark across 100 replications. IntervalGP-VAE achieves lower or comparable PEHE and ATE error to strong baselines (e.g., TEDVAE Zhang et al. (2021)) while additionally providing calibrated ITE intervals.

## 2 PROBLEM SETTING

Key notations used in this paper are listed in Table 1 for clarity and brevity. We assume the outcome is generated according to the following structural equation:

$$Y = f(T, U, Z_Y) + \epsilon_Y \qquad (1)$$

where $f : \{0,1\} \times \mathbb{R}^d \times \mathbb{R}^{k'} \to \mathbb{R}$ is a potentially nonlinear function and $Z_Y \subseteq Z \in \mathbb{R}^k$ represents a selected subset of observed proxy variables that may directly influence the outcome (e.g., acting as mediators or additional covariates), with $k' < k$. The noise term $\epsilon_Y$ is assumed exogenous, satisfying $\epsilon_Y \perp (T, U, Z_Y)$. We observe samples of the triplet $(Z, T, Y)$, where $Z$ provides indirect information about the latent confounder $U$, with its informativeness determined by the structural assumptions. Each proxy variable $Z_i$ is assumed to be generated from $U$ via a noisy, smooth, and injective function:

Table 1: Summary of key notations.

| Symbol | Description |
|---|---|
| $Z \in \mathbb{R}^k$ | Proxy variables |
| $Z_Y \in \mathbb{R}^{k'}$ | Auxiliary variables |
| $T \in \{0,1\}$ | Binary treatment variable |
| $Y \in \mathbb{R}$ | Outcome variable |
| $U \in \mathbb{R}^d$ | Latent confounder(s) |
| $\epsilon_Y, \epsilon_Z$ | Noise for $Y$ and $Z$ |
| $f(T, U, Z_Y)$ | Outcome function |
| $g(U)$ | Mapping from $U$ to proxies |
| $\hat{Y}(t)$ | Estimated outcome under $t$ |
| $q(u \mid z)$ | Posterior over $U$ (encoder) |
| $p_\theta(z \mid u, \epsilon)$ | Proxy likelihood (decoder) |
| $[\hat{\tau}_j^{\text{lower}}, \hat{\tau}_j^{\text{upper}}]$ | ITE interval for individual $j$ |

$$Z_i = g_i(U) + \epsilon_{Z_i}, \quad \epsilon_{Z_i} \perp U, \quad i = 1, \dots, k, \qquad (2)$$

where $\epsilon_{Z_i}$ are mutually independent noise terms, and the functions $g_i : \mathbb{R} \to \mathbb{R}$ are unknown but sufficiently non-redundant. Given observed samples $\{(Z^i, T^i, Y^i)\}_{i=1}^n$, the objective is to understand the identifiability of the latent confounder $U$ and its implications for recovering downstream causal quantities. Specifically, we aim to address:

- **Identifiability:** Under what structural or statistical conditions can the latent variable $U$ be recovered (up to an equivalence class such as an invertible or monotonic transformation), and treatment effect $\tau(U) := f(1, U, Z_Y) - f(0, U, Z_Y)$ becomes identifiable.
- **Quantification of uncertainty:** When $U$ is only partially identifiable, we aim to derive meaningful *bounds* (e.g., lower and upper bounds) on treatment outcomes and effects, and characterize these bounds via confidence intervals or posterior uncertainty regions whenever feasible.

To support the identifiability, we adopt the following standard assumptions from causal inference:

**(i) Positivity:** Every individual has a non-zero probability of receiving both treatment and control, i.e.,

$$0 < P(T = 1 \mid U) < 1. \tag{3}$$

**(ii) Latent Ignorability:** Treatment assignment is independent of potential outcomes conditional on $U$:

$$(Y(0), Y(1)) \perp\!\!\!\perp T \mid U. \tag{4}$$

This implies that $U$ fully accounts for confounding between between $T$ and $Y$, and that $Z_Y$ does not introduce spurious confounding.

**(iii) Consistency and Well-Defined Outcomes:** For each treatment level $t \in \{0, 1\}$, the potential outcome $Y(t)$ is generated by the structural equation

$$Y(t) = f(t, U, Z_Y) + \varepsilon_Y, \tag{5}$$
$$\varepsilon_Y \perp (T, U, Z_Y). \tag{6}$$

Consistency holds in the sense that if $T = t$ then $Y = Y(t)$. Hence, outcomes are well-defined as functions of the latent confounder $U$ and auxiliary covariates $Z_Y$.

A common assumption in prior proxy-based models (e.g. CEVAE (Louizos et al., 2017)) is that the full proxy vector $Z$ is conditionally independent of both treatment and outcome given $U$: $Z \perp\!\!\!\perp (T, Y) \mid U$. However, this assumption is often unrealistic in practice. Our formulation relaxes this constraint to capture richer and more realistic causal structures.

## 3 RELATED WORK

We categorize related work into three areas: latent confounder modeling, Gaussian Processes and VAE, and interval-valued Gaussian Processes.

### 3.1 FROM PROXY-BASED IDENTIFICATION TO PERSONALIZED LATENT RECOVERY

Leveraging proxy variables to identify causal effects in the presence of unobserved confounding is a foundational strategy in causal inference. Early work by Kuroki and Pearl Kuroki & Pearl (2014) and Miao et al. Miao et al. (2018) established identifiability conditions using proxies under linear and parametric assumptions. More recent approaches, such as Deep Proxy Causal Learning (Deep-PCL) Xu et al. (2021), extend these ideas to nonlinear settings via neural architectures. However, these methods typically focus on population-level identification. Also, some of them Xu et al. (2021) rely on unverifiable assumptions, such as partitioning proxies into treatment- or outcome-specific subsets. There is growing interest in *personalized* latent recovery, particularly via nonlinear ICA Hyvärinen et al. (2019) and its variational counterpart iVAE Khemakhem et al. (2020). In causal inference, VAE-based models such as CEVAE Louizos et al. (2017), TEDVAE Zhang et al. (2021), CEMVAE (Wu et al., 2024), InfoVAE Zhao et al. (2019) and InfoCEVAE Harada & Kashima (2024) have been proposed to learn latent confounders from proxies. Gaussian Process-based alternatives include the Sequential Deconfounder Kuzmanovic et al. (2021); Hatt & Feuerriegel (2024) and Structured GP Confounder Witty et al. (2020). However, most of these approaches lack structural identifiability guarantees. InfoVAE Zhao et al. (2019) and InfoCEVAE Harada & Kashima (2024) encourages latent recovery via mutual information, but does not provide formal identifiability analysis linking proxies to latent variables. Our method departs from prior work by establishing structural identifiability through the tensor decomposition framework of Allman et al. Allman et al. (2009). We show that under nonlinear, injective proxy mappings, the latent confounder $U \in \mathbb{R}^d$ can be recovered (up to a smooth invertible transformation) from $k \geq 2d + 1$ proxy variables, where $d$ is the number of latent confounders. This result ensures that the recovered latent representation supports valid causal inference, including ITE estimation. Moreover, our framework generalizes a wide range of causal structures involving latent confounders and proxy variables, including but not limited to, the settings addressed by CEVAE and TEDVAE. It enables recovery of latent confounders from a unified proxy set under a general DAG-based formulation. This flexibility allows the model to accommodate diverse proxy–treatment–outcome dependency structures within a single principled and identifiable framework.

## 3.2 GP-VAE AND UNCERTAINTY-AWARE ESTIMATION

Gaussian Processes (GPs) offer a flexible, nonparametric framework for uncertainty modeling (Rasmussen & Williams, 2006). The foundational work by (Casale et al., 2018) introduced GP-VAEs, replacing the isotropic Gaussian prior in VAEs with a GP prior to induce structured latent representations that vary smoothly with the input. Subsequent work extended this idea to sequential data (Fortuin et al., 2020) and latent confounder trajectories (Hatt & Feuerriegel, 2024). In the Structured GP Confounder model Witty et al. (2020), separate GPs are used to model treatment and outcome given latent confounders. Our proposed IntervalGP-VAE introduces an interval-valued GP prior over the latent space to capture both structured dependencies and epistemic uncertainty in confounder recovery from noisy proxies. For outcome prediction, we incorporate a GP head trained on interval-valued targets to produces calibrated calibrated predictive uncertainty for counterfactual queries. To our knowledge, this is the first model to integrate GP-based latent inference with interval-valued uncertainty propagation for personalized treatment effect estimation.

## 3.3 INTERVAL-VALUED GAUSSIAN PROCESSES

Modeling interval-valued outputs with GPs has received limited attention in the literatures. To the best of our knowledge, the only existing work that explicitly supports interval observations in a GP setting is the Generalized Multi-Output Censored GP model by Gammelli et al. (2020; 2022). Their framework introduces a likelihood formulation capable of handling output intervals across multiple outputs. However, their method is designed for multi-output regression tasks and does not address causal inference, latent confounders, or proxy variables. Our approach supports interval supervision of the latent space and, through a GP, interval prediction of individual treatment effects. This allows us to model both uncertainty in confounder inference and outcome prediction in a principled and calibrated manner.

## 4 IDENTIFIABILITY ANALYSIS

**Definition 1** (Identifiability). *A latent variable $U$ is said to be* identifiable *if there exists a mapping from the observed variables to $U$, up to a smooth and invertible transformation, such that model outputs, e.g., counterfactual outcomes or treatment effects, remain invariant under that transformation.*

If the latent confounder $U$ were fully observed, the structural outcome function in equation 1 would yield identifiable causal effects under standard assumptions. However, when $U$ is unobserved and must be inferred from proxies, the key question is: under what conditions is $U$ still identifiable, and valid causal effects recoverable? Theorem 1 formally characterizes the identifiability conditions necessary for such recovery.

*Proof.* See Appendix A for the detailed proof of Theorem 1, and Appendix B for an illustrative example.     □

**Theorem 1** (Identifiability of a Latent Variable from Noisy Proxy Variables)

Under the proxy structural equation denoted by equation 2, suppose the latent confounder $U$ generates $k$ proxy variables $Z_i$, where each $g_i : \mathbb{R}^d \to \mathbb{R}$ is unknown, continuously differentiable, and injective, and the noise terms $\epsilon_{Z_i}$ are mutually independent and independent of $U$. Then:

- If $k < 2d + 1$, then $U$ is not identifiable from the marginal distribution $p(Z)$ in general.

- If $k \geq 2d + 1$, and the functions $\{g_i\}_{i=1}^{k}$ are sufficiently smooth, nonlinear, and non-redundant, then $U$ is identifiable from $p(Z)$, up to a smooth and invertible transformation.

The identifiability result in Theorem 1 is based on the following assumptions: 1) Structural Form: The observed proxies $Z_i$ are generated from the latent confounder $U$ via the structural equations equation 2, where the additive noise terms $\epsilon_{Z_i}$ are mutually independent and independent of $U$. The functions $\{g_i\}$ are not only smooth, injective, but also sufficiently nonlinear and non-redundant, meaning they provide diverse and informative mappings of $U$. 2) Conditional Independence of Proxies: The proxy variables $Z_i$ are conditionally independent given $U$, i.e.,

$$Z_1 \per\!\!\!\perp Z_2 \per\!\!\!\perp \cdots \perp\!\!\!\perp Z_k \mid U.$$

Since identifiability only holds up to a smooth, strictly monotonic, and invertible transformation of $U$, a natural question arises: does this ambiguity affect ITE estimation? Theorem 2 establishes that ITE estimation is invariant to smooth, monotonic, and invertible reparameterizations of the latent space. This result holds for fixed $Z_Y$[1].

---

**Theorem 2** (Invariance of the ITE under Transformations of the Latent Space)

Let $h : \mathbb{R} \to \mathbb{R}$ be a smooth, strictly monotonic, and invertible function. Then the individual treatment effect (ITE) remains invariant under such transformations of the latent confounder. Specifically, if $\hat{U} = h(U)$, then $\text{ITE}(\hat{U}_i) = \text{ITE}(U_i)$.

---

*Proof.* See the detailed proof of Theorem 2 in Appendix D. □

## 5 INTERVALGP-VAE

### 5.1 MOTIVATION FOR GP PRIOR

---

**Proposition 1** (GP Priors Enable Regularized and Invariant Latent Recovery in Causal Models)

Let $U = (u_1, \ldots, u_n)^\top$ denote latent confounder values for $n$ observed samples, each associated with proxy observations $Z^i \in \mathbb{R}$ via the structural equation equation 2. Suppose a Gaussian Process prior is placed over $U$ as

$$U \sim \mathcal{GP}(0, K(Z^i, Z^j)),$$

where $K$ is a smooth, positive-definite kernel over the proxy space. Then:

1. The GP prior does not affect the identifiability of causal effects, such as ITEs.

2. For any smooth and invertible transformation $h : \mathbb{R} \to \mathbb{R}$, the transformed latent $\tilde{U} = h(U)$ inherits the same geometric structure via an induced kernel, preserving regularization.

---

*Proof.* See Appendix E for the detailed proof of Proposition 1. □

Proposition 1 holds under the following additional assumptions: 1) the kernel function $K(Z^i, Z^j)$ is smooth and positive-definite; 2) the GP prior encodes a distribution over $U$ that respects relative similarity in the proxy space without enforcing absolute coordinates; and 3) identifiability of $U$ is defined up to a smooth, strictly monotonic, and invertible transformation, consistent with Theorem 2.

### 5.2 INTERVALGPS FOR LATENT REPRESENTATION AND ITE ESTIMATION

We extend the Variational Autoencoder (VAE) framework by introducing a structured prior over latent variables using *Interval Gaussian Processes (IntervalGP)* Gammelli et al. (2020; 2022). Unlike standard VAEs that impose an isotropic Gaussian prior over latent representations, we define a GP prior over the proxy space $Z$ and model the latent variable $U$ as an interval-valued random function of $Z$.

**Interval-Valued GP Regression.** Given input features $X^i \in \mathbb{R}^d$ and scalar targets $y^i \in \mathbb{R}$, standard GP regression assumes a latent function $f(\mathbf{x}) \sim \mathcal{GP}(0, K(X, X'))$, where $K(\cdot, \cdot)$ is the kernel:

$$k(X^i, X^j) = \sigma_f^2 \exp\left(-\frac{\|X^i - X^j\|^2}{2\ell^2}\right), \tag{7}$$

---

[1]If $Z_Y$ appears in the outcome model, it is assumed to be observed and held fixed when evaluating counterfactual outcomes.

with hyperparameters $\sigma_f^2$ (variance) and $\ell$(lengthscale). In many practical settings, however, targets are not point-valued but known to lie within intervals: $y^i \in \left[y^{\text{lower}, i}, y^{\text{upper}, i}\right]$. To accommodate this, IntervalGP Gammelli et al. (2020; 2022) generalizes standard GP regression to interval targets by replacing the Gaussian likelihood with a truncated Gaussian likelihood:

$$p\big(y^{\text{lower},i} \leq f(X^i) \leq y^{\text{upper},i} \mid X^i\big) = \Phi\bigg(\frac{y^{\text{upper},i} - \mu^i}{\sigma^i}\bigg) - \Phi\bigg(\frac{y^{\text{lower},i} - \mu^i}{\sigma^i}\bigg). \tag{8}$$

where $\mu^i$ and $\sigma^i$ denote the predictive mean and standard deviation of the GP at $X^i$, and $\Phi(\cdot)$ is the standard Gaussian cumulative distribution function (CDF).

**IntervalGP Prior on the Latent Confounder.** To infer the latent confounder $U$ from observed proxies $Z$, we employ a variational encoder:

$$q(U \mid Z) = \mathcal{N}(\mu_u(Z), \sigma_u^2(Z)), \tag{9}$$

where each latent estimate $U_i$ is treated as an interval-valued random variable, interpreted as $u_i \in [\mu_u^i - \sigma_u^i, \ \mu_u^i + \sigma_u^i]$.

To regularize the learned latent space $U$, we impose a Gaussian Process prior over $U(Z)$, i.e., $U(Z) \sim \mathcal{GP}(0, \ k(Z, Z'))$. is placed over the latent space to regularize the learned $U$-space. This GP prior treats each encoder-derived interval as an interval-valued observation under the GP posterior, enabling structured and uncertainty-aware regularization. For each data point $i$:

$$\log p_{\text{GP}}\big(u^i \in [\mu_u^i \pm \sigma_u^i] \mid Z^i\big) = \log\bigg[\Phi\bigg(\frac{\mu_u^i + \sigma_u^i - \mu^{\text{GP},i}}{\sigma^{\text{GP},i}}\bigg) - \Phi\bigg(\frac{\mu_u^i - \sigma_u^i - \mu^{\text{GP},i}}{\sigma^{\text{GP},i}}\bigg)\bigg]. \tag{10}$$

where $\mu^{\text{GP}, i}, \sigma^{\text{GP}, i}$ denote the posterior predictive mean and standard deviation of the GP at input $Z^i$.

**Latent Confounder Prediction via GP Posterior.** The GP prior enables coherent prediction for unseen inputs using interval-valued observations. Specifically, we define:

- $K \in \mathbb{R}^{n \times n}$: the kernel matrix over training inputs, with $K_{ij} = K(Z^i, Z^j) + \sigma^2 \delta^{ij}$,
- $K_s \in \mathbb{R}^{n \times 1}$: the cross-covariance vector between training inputs and a test input $Z^*$,
- $K_{ss} \in \mathbb{R}$: the prior variance at $Z^*$, i.e., $K_{ss} = k(Z^*, Z^*) + \sigma^2$.

The GP posterior over the latent confounder at $Z^*$ is:

$$p(u^* \mid Z^*, \mathcal{D}) = \mathcal{N}(\mu^*, \sigma^{*2}), \tag{11}$$

where $\mu_u^* = K_s^\top K^{-1} \mu_u$, $\sigma_u^{*2} = K_{ss} - K_s^\top K^{-1} K_s$. The resulting latent interval is interpreted as: $u^* \in [\mu_u^* - \sigma_u^*, \ \mu_u^* + \sigma_u^*]$.

**ITE Prediction via GP Posterior.** Similarly, the GP posterior for the ITE at test input $Z^*$ is computed as:

$$\mu_{\text{ITE}}^{\text{lower}}(Z^*) = K_s^\top K^{-1} \mu_{\text{ITE}}^{\text{lower}}, \quad \mu_{\text{ITE}}^{\text{upper}}(Z^*) = K_s^\top K^{-1} \mu_{\text{ITE}}^{\text{upper}}, \quad \sigma_{\text{ITE}}^2(Z^*) = K_{ss} - K_s^\top K^{-1} K_s. \tag{12}$$

where $\mu_{\text{ITE}}^{\text{lower}}$ and $\mu_{\text{ITE}}^{\text{upper}}$ are obtained by drawing multiple samples from the GP posterior $q(u \mid Z^*)$ and computing empirical quantiles:

$$\text{ITE}_{\text{lower}} = \text{Quantile}_\alpha(Y_1 - Y_0), \quad \text{ITE}_{\text{upper}} = \text{Quantile}_{1-\alpha}(Y_1 - Y_0). \tag{13}$$

The final ITE prediction is expressed as an interval: $\text{ITE}_{\text{GP}}(Z^*) \in [\mu_{\text{ITE}}^{\text{lower}}(Z^*), \ \mu_{\text{ITE}}^{\text{upper}}(Z^*)]$, with uncertainty quantified by the predictive variance $\sigma_{\text{ITE}}^2(Z^*)$.

## 5.3 INTERVALGP-VAE ARCHITECTURE

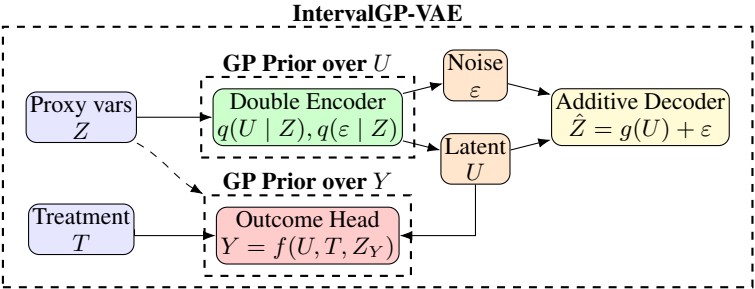

Figure 1: Architecture of the **IntervalGP-VAE** model.

The proposed *IntervalGP-VAE* extends the standard VAE framework by incorporating structured GP priors over both the latent confounder space $U$ and ITE. This architecture enables the model to produce both point estimates and calibrated uncertainty intervals for latent variables and causal effects. The key components are outlined below and illustrated in Fig. 1: Encoder: Maps each input $Z^i$ to a variational posterior over $U$, defining a latent interval $u^i \in [\mu_u^i - \sigma_u^i, \ \mu_u^i + \sigma_u^i]$ via equation 9 to capture epistemic uncertainty in the latent representation. IntervalGP over $U$: A GP prior regularizes the latent space by maximizing the interval-based log-likelihood $\log p_{\text{GP}}(\mu_u \pm \sigma_u \mid Z)$ as given in equation 10. Decoder: Reconstructs the proxies via $\hat{\mathbf{z}} = g_{\text{dec}}(u, \epsilon)$, using sampled latent variables and noise. Outcome Head: Predicts the outcome $y$ from $u, t$, enabling estimation of potential outcomes and corresponding ITEs. IntervalGP over ITE: A second GP regressor predicts calibrated ITE intervals, propagating uncertainty from latent inference to treatment effect estimation. To train the model, we formulate a

---

**Algorithm 1:** IntervalGP–VAE Algorithm

**Input:** Train set $\mathcal{D}_{\text{train}}$, test set $\mathcal{D}_{\text{test}}$, joint training epochs $E$

**Output:** Estimated ITEs $\{\hat{\tau}_j\}_{j=1}^m$; ITE intervals $[\hat{\tau}_j^{\text{lower}}, \hat{\tau}_j^{\text{upper}}]$

**Initialize** model $\mathcal{M}$ with encoder $q_\phi(u, \epsilon \mid z)$, decoder $p_\theta(z \mid u, \epsilon)$, and causal head $f_\psi(u, t)$;

**for** epoch $= 1$ **to** $E$ **do**
    **for** *each mini-batch* $(z, t, y)$ **do**
        Sample $(u, \epsilon) \sim q_\phi(u, \epsilon \mid z)$;
        Reconstruct $\hat{z} \sim p_\theta(z \mid u, \epsilon)$;
        Predict $\hat{y} = f_\psi(u, t, z_y)$;
        Compute total loss $\mathcal{L}$ and update $(\phi, \theta, \psi)$;

**Posterior over** $u$: for each $z_j$ estimate $q_\phi(u \mid z_j)$ and draw $u_j^{(s)}$;

Estimate $y_j(0), y_j(1)$, compute ITE and CI $[\text{ITE}_j^{\text{lower}}, \text{ITE}_j^{\text{upper}}]$;

Fit GP (RBF) $z_{\text{train}} \to u_{\text{train}}$; predict $u_j$ for test $z_j$;

Compute $\hat{y}_j(0), \hat{y}_j(1)$ and $\text{ITE}_j = \hat{y}_j(1) - \hat{y}_j(0)$; obtain GP-based ITE intervals;

---

composite training objective that jointly optimizes reconstruction accuracy and GP-based prior regularization. The overall training procedure is summarized in Algorithm 1. The weights and implementation-specific parameters used in the IntervalGP-VAE method are detailed in the experiment section.

## 6 EXPERIMENTS

We conduct both synthetic and semi-synthetic experiments to evaluate the effectiveness of our proposed method. All experiments were conducted on a laptop running *Windows 11 Home* (version 22H2, build 22631), equipped with a 13th Gen Intel® Core™ i9-13900H processor (14 cores, 20 threads, 2.6 GHz), 32 GB of RAM, and a 1 TB SSD. We compare our **IntervalGP-VAE** method with **TEDVAE** (Zhang et al., 2021), which outperforms a range of state-of-the-art methods. These include traditional approaches such as the Squared t-statistic Tree (t-stats) (Su et al., 2009) and Causal Tree (CT) (Athey & Imbens, 2016); ensemble-based methods such as Causal Random Forest (CRF) (Wager & Athey, 2018), Bayesian Additive Regression Trees (BART) (Hill, 2011), and the X-Learner (Künzel et al., 2019) with Random Forest (Breiman et al., 1984) as the base learner (X-RF); deep representation learning methods including Counterfactual Regression Net (CFR) (Shalit et al., 2017), Similarity Preserved Individual Treatment Effect (SITE) (Yao et al., 2018), and the variable decomposition method DR-CFR (Hassanpour & Greiner, 2020); as well as generative approaches such as the Causal Effect Variational Autoencoder (CEVAE) (Louizos et al., 2017) and

GANITE (Yoon et al., 2018). We evaluate model performance using two standard metrics: Precision in Estimation of Heterogeneous Effect (PEHE) and Average Treatment Effect (ATE) error (Hill, 2011; Shalit et al., 2017; Louizos et al., 2017; Yao et al., 2018). In addition, we also report the coverage rate of the estimated individual treatment effect intervals, a distinguishing feature of our method that quantifies its ability to capture uncertainty in counterfactual predictions. Model training follows a staged strategy: joint training of the encoder, decoder, and causal head for $E = 200$ epochs using the Adam optimizer with a batch size of 128 and learning rate of $10^{-3}$. The implementation of IntervalGP-VAE uses the following parameters: latent dimension is 1, hidden layer width is 64, GP lengthscale $\ell = 0.4$, GP variance $\sigma_f^2 = 5.0$, and GP noise variance $= 10^{-4}$.

## 6.1 Synthetic Experiments

Table 2: Treatment mechanisms and proxy/outcome functions used in synthetic experiments.

| | Functions |
|---|---|
| **Proxy Functions** | $\{u, \sin(u), u^2\}, \{\tanh(u), \sin(2u), \log(\|u\| + 10^{-3})\}, \{u^2 + u_{a0}, \log(1 + \|u\|) + u_{a1},$ $u^3 + 0.1 \cdot u_{a0} \cdot u_{a1}\}$ $\{\tanh(u) + u_{a0}, \arctan(u) + 0.1 \cdot u_{a1}, \sin(u) + \exp(-\|u_{a0}\|) + u\}$ $\{\frac{u}{\|u_{a0}\|+0.1}, \frac{\sin(u)}{1+u^2} + 0.05\,u_{a1}, \log(1 + u^2) + 0.2\,u_{a0}\}$ $\{\log(1 + e^u) + 0.1\,u_{a0}, u^3 + 0.1\,u_{a1}, \sigma(u) + 0.05\,u_{a0}u_{a1}\}$ |
| **Treatment Functions** | $\{\text{Bernoulli}(\sigma(1.5u + 0.8u_{a0}))\}, \{\text{Bernoulli}(\sigma(0.5u + 1.2u_{a0}))\}$ |
| **Outcome Functions** | $\{\sin(u) + u^2 + 0.3\,u_{a0} + 0.3\cos(u_{a1}) + \varepsilon\}, \{\sin(u) + t + 0.5ut + 0.5\cos(u_{a1}) + \epsilon\}$ |

We evaluate the methods on 24 synthetic settings combining proxy functions, binary treatment mechanisms, and outcome functions, as detailed in Table 2. The proxy functions are constructed to satisfy the identifiability conditions outlined in Theorem 1. Specifically, three proxy functions are used—the minimal number required to identify a single latent confounder according to Theorem 1. This experimental setup is designed to validate the effectiveness of the proposed IntervalGP-VAE framework under theoretically justified conditions. For each setting, we generate 1,000 training samples and 50 testing samples. The noise in the outcome function is drawn from a normal distribution with standard deviation $\sigma = 0.1$, and the latent confounder $u$ is sampled from a standard normal distribution. The results are presented in Figure 3a. From Figure 3a, we can observe that, *IntervalGP-VAE* achieves a lower average PEHE (orange dashed line) compared to *TEDVAE* (blue dashed line), while attaining comparable ATE error. Notably, *IntervalGP-VAE* exhibits a high coverage rate exceeding 90% (91.9%), indicating that its predicted ITE intervals are well-calibrated with respect to the 90% confidence level. These findings validate the theoretical claims of the paper: when the proxy functions satisfy the identifiability conditions formalized in Theorem 1, the latent confounder $U$ becomes recoverable up to equivalence, and the model can accurately quantify uncertainty in the inferred treatment effects. Figure 2 displays the Gaussian-process (GP) posterior intervals for the ITE on one of the 24 synthetic replicates, computed by conditioning the GP on five randomly sampled training points (highlighted in pink) to enforce smoothness. The figure indicates that the learnt latent structure yields well-calibrated uncertainty quantification for individualized treatment effects.

## 6.2 Semi-synthetic Experiments

Beyond the synthetic settings, we conduct a 100-replication study on the semi-synthetic IHDP benchmark (Hill, 2011). In this setting, the observed covariates $Z$ are treated as proxies for latent socio-demographic confounders $U$, providing a realistic yet evaluable testbed with ground-truth counterfactuals. To reduce variance and ensure reproducibility, we use only the continuous covariates specified by the IHDP in-

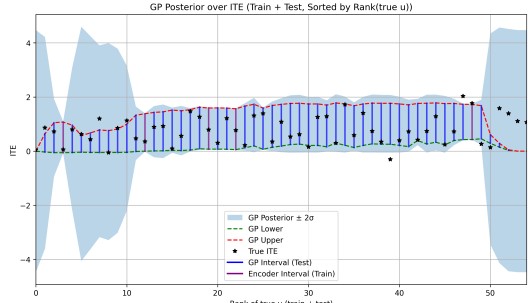

Figure 2: GP posterior ITE intervals.

dex. We follow the standard evaluation protocol: train on the prescribed training split and assess on the held-out test split across multiple realizations, reporting PEHE, ATE error, and the empirical coverage of the 90% confidence intervals. The results are presented in Figure 3b. On the IHDP benchmark, IntervalGP-VAE achieves comparable performance in terms of PEHE and ATE error,

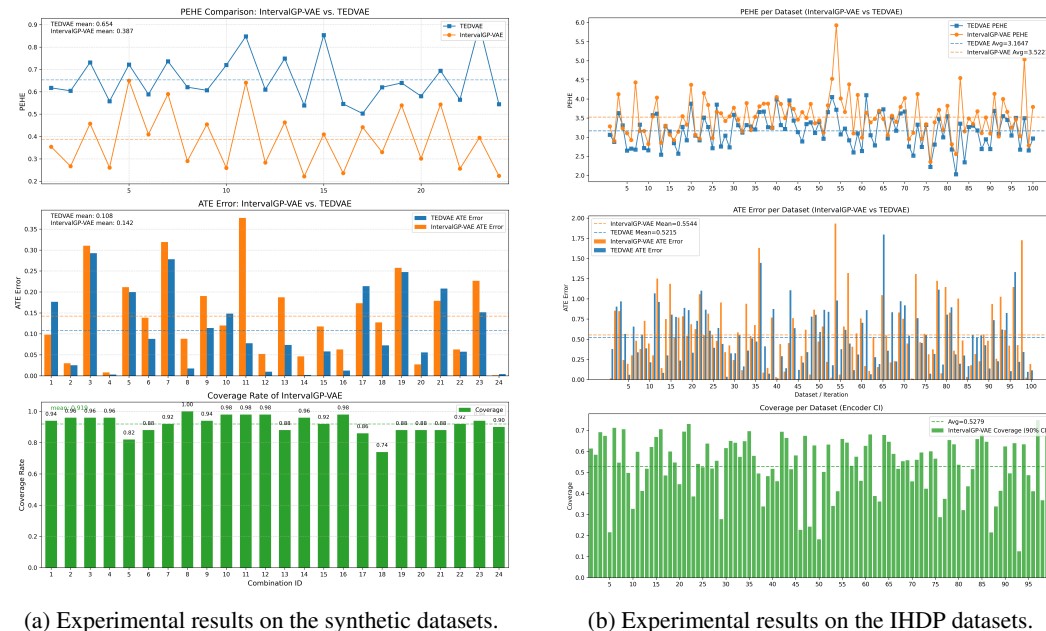

(a) Experimental results on the synthetic datasets.  (b) Experimental results on the IHDP datasets.

Figure 3: Experimental results on synthetic (left) and IHDP (right) datasets.

relative to TEDVAE. However, the empirical coverage rates of the 90% confidence intervals are lower than those observed on the synthetic datasets (52.8% vs 91.9%). This discrepancy can be theoretically explained by the identifiability conditions outlined in Theorem 1. Specifically, the synthetic datasets are constructed to satisfy the minimum sufficient conditions for identifying the latent confounder via multiple proxy variables. In contrast, the proxy generation mechanism in the IHDP dataset may fail to fulfil all these conditions, such as the use of of at least three nonlinear, complementary proxy functions, thereby weakening the model's ability to reliably infer the true latent structure. Additionally, the GP prior in IntervalGP-VAE assumes that the latent variable is a smooth function of observed proxies. If this assumption is violated, due to poor alignment between the selected covariates and the underlying confounder structure, both posterior inference and interval calibration may degrade. Furthermore, the performance sensitivity to GP hyperparameters, such as the prior variance $\sigma_f^2$ and lengthscale $\ell$, becomes more pronounced in real-world settings where proxy informativeness is limited. This highlights the practical importance of model selection and proxy variable design when applying theory-grounded causal inference frameworks such as IntervalGP-VAE to observational datasets. Therefore, relaxing and extending the identifiability conditions underpinning IntervalGP-VAE becomes a critical direction for future work.

## 7 CONCLUSION

We presented IntervalGP-VAE, a generative framework for estimating individualized treatment effects (ITEs) under unobserved confounding. By disentangling latent confounders and measurement noise from noisy proxies and imposing an interval-valued Gaussian process prior, the model provides well-calibrated ITE intervals rather than only point estimates. Our identifiability analysis shows that, under minimal structural conditions on the proxy generation mechanism, the latent confounder can be recovered up to a smooth monotonic transformation and that ITE estimation remains invariant to such transformations, as demonstrated on synthetic and semi-synthetic benchmarks. Future directions include: first, integrating richer latent structures, including multivariate or hierarchical confounders, with deeper theoretical guarantees. Second, temporal or spatial extensions would enable counterfactual reasoning in longitudinal health, education, or environmental studies. Third, robustness to incomplete or partially informative proxies, via adaptive kernel learning or causal feature selection, would enhance applicability to real-world observational data. Finally, exploring decision-theoretic uses of ITE intervals, such as risk-sensitive treatment recommendation and fairness-aware policy optimization, could enhance the societal impact of uncertainty-aware causal inference.

## ETHICS STATEMENT

This work adheres to the ICLR Code of Ethics . The proposed IntervalGP-VAE framework is developed for methodological research in causal inference and does not involve human subjects, personally identifiable information, or sensitive attributes. All datasets used are either synthetic or the publicly available IHDP benchmark, which is fully de-identified and commonly used in causal inference research. No proprietary or restricted-access data were employed. The methods and results do not promote harmful applications or discriminatory practices, and there are no known risks to privacy, security, or safety. All theoretical claims are supported by complete proofs in the supplementary materials, ensuring research integrity and transparency. We disclose no conflicts of interest or sponsorship that could influence this research.

## REPRODUCIBILITY STATEMENT

To ensure reproducibility of the results presented in Fig. 3, the theoretical assumptions and identifiability guarantees underpinning the model are stated in Sec. 4 and proven in Apps. B–F. All modeling details—including the IntervalGP-VAE architecture and the complete training procedure—are provided in Sec. 5. The synthetic data-generating mechanism settings are enumerated in Table 2. Hyperparameters and experimental setups for both synthetic and semi-synthetic (IHDP) datasets are described in Sec. 6. To facilitate exact replication, an anonymized implementation of IntervalGP-VAE together with data-processing scripts and configuration files are submitted as supplementary material.

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

# Appendix

## A   LLM USAGE DISCLOSURE

In this work, Large language models (LLMs) were used as an auxiliary tool to assist refine statements and improve grammar, style, and readability of the written text. All conceptual development, technical derivations, experiments, and final claims are the authors' own work, and the authors take full responsibility for the correctness and originality of the content.

## B   PROOF OF THEOREM 1

---

**Theorem 1** (Identifiability of a Latent Variable from Noisy Proxy Variables)

Under the proxy structural equation defined in equation 2, assume the latent confounder $U$ generates $k$ observed proxy variables $Z_i$, where each $g_i : \mathbb{R}^d \to \mathbb{R}$ is unknown, continuously differentiable, and injective, and the noise terms $\epsilon_i$ are mutually independent and independent of $U$. Then:

- If $k < 2d + 1$, then $U$ is not identifiable from the marginal distribution $p(Z)$, in general.
- If $k \geq 2d + 1$, and the functions $\{g_i\}_{i=1}^k$ are sufficiently smooth, nonlinear, and non-redundant, then $U$ is identifiable from $p(Z)$, up to a smooth and invertible transformation.

---

*Proof.* Let $p(Z \mid U) = \prod_{i=1}^k p(Z_i \mid U)$, where each $p(Z_i \mid U)$ is induced by a transformation of the noise $\epsilon_i$ through the mapping in equation 2. The joint distribution $p(Z \mid U)$ defines a smooth manifold over $U \in \mathbb{R}^d$. The marginal distribution over $Z \in \mathbb{R}^k$ can be written as:

$$p(Z) = \int p(Z \mid U) \, p(U) \, dU. \tag{14}$$

To apply identifiability results from tensor decomposition theory (e.g., Kruskal's theorem via Allman et al. (2009)), we temporarily discretize each observed proxy $Z_i$ into $n_i$ bins (e.g., via quantization or histogram binning). This induces a discrete representation of the joint distribution $p(Z \mid U)$, which can be grouped into three disjoint subsets $L_1, L_2, L_3 \subseteq \{1, \ldots, k\}$ with $k_1 + k_2 + k_3 = k$.

Let $r$ denote the number of discretized latent bins for $U$. Then for each group $L_j$, we construct the conditional matrix:

$$M_j = \begin{bmatrix} P(z_1^{(L_j)} \mid u_1) & \cdots & P(z_{n_j}^{(L_j)} \mid u_1) \\ \vdots & \ddots & \vdots \\ P(z_1^{(L_j)} \mid u_r) & \cdots & P(z_{n_j}^{(L_j)} \mid u_r) \end{bmatrix} \in \mathbb{R}^{r \times n_j}, \quad \text{for } j = 1, 2, 3. \tag{15}$$

These matrices represent the discretized conditional distributions of grouped proxies given $U$. The joint distribution over $Z$ in discretized space induces a 3-way tensor $T \in \mathbb{R}^{n_1 \times n_2 \times n_3}$ with a CP (PARAFAC) representation

$$T = [M_1, M_2, M_3]_{\text{CP}} = \sum_{\ell=1}^r a_\ell \otimes b_\ell \otimes c_\ell, \tag{16}$$

where the factor matrices are $M_1, M_2, M_3$. Let $L_1, L_2, L_3$ be a *partition* of $\{Z_1, \ldots, Z_k\}$ (pairwise disjoint and $L_1 \cup L_2 \cup L_3 = \{Z_1, \ldots, Z_k\}$). Under conditional independence of proxies given $U$ and sufficient variability (generic position) within each group, the Kruskal ranks satisfy

$$k(M_j) \geq \min\{|L_j|, r\}, \qquad j = 1, 2, 3. \tag{17}$$

Hence

$$k(M_1) + k(M_2) + k(M_3) \geq \min\{|L_1|, r\} + \min\{|L_2|, r\} + \min\{|L_3|, r\}$$
$$\geq \min\{k, 2r + 2\}. \tag{18}$$

By Kruskal's condition,

$$k(M_1) + k(M_2) + k(M_3) \; \geq \; 2r + 2, \tag{19}$$

which is guaranteed, for example, when $k \geq 2r + 2$ and each $|L_j| \leq r$ (or, more generally, whenever $\min\{|L_1|, r\} + \min\{|L_2|, r\} + \min\{|L_3|, r\} \geq 2r + 2$). Taking $r \geq d + 1$ (so that the discretization retains at least $d + 1$ latent states) yields the sufficient requirement

$$k \geq 2r + 2 \; \geq \; 2d + 4. \tag{20}$$

Kruskal's theorem guarantees that the decomposition of $T = [M_1, M_2, M_3]$ is unique up to simultaneous row permutation and scaling. Thus, the prior $p(U)$ and the conditional densities $p(Z \mid U)$ are identified (up to permutation). Then from Bayes' theorem we have:

$$p(U \mid Z) = \frac{p(Z \mid U)\, p(U)}{p(Z)}, \tag{21}$$

Thus, the posterior distribution $p(U \mid Z)$ is identifiable.

$\square$

Note that, the discretization is only used as a proof device; our result does not depend on the particular binning scheme or discretization level. As the bin widths shrink, the discrete model approaches the continuous distribution. Therefore, identifiability in the discrete approximation implies identifiability in the original continuous setting, by standard arguments of approximation and continuity of probability densities. See a toy illustration of tensor construction from conditional matrices in Appendix C.

## C  AN ILLUSTRATIVE EXAMPLE FOR THEOREM 1

Let us consider cases when $d = 1$ as below:

CASE 1: $k = 1$ — NOT IDENTIFIABLE

Consider the following proxy variable:

$$Z_1 = g_1(u) = \tanh(u),$$

where the function $g_1$ is smooth, strictly increasing, and injective on $\mathbb{R}$. Now, consider a smooth, strictly monotonic transformation $h(u) = u + 2$, so that $v = h(u)$ and $u = h^{-1}(v) = v - 2$. Then, define:

$$\tilde{g}_1(v) := g_1(h^{-1}(v)) = \tanh(v - 2),$$

and observe that:

$$Z_1 = g_1(u) = \tilde{g}_1(v).$$

This shows that the same observed value $Z_1$ could have been produced from either $u$ or $v = h(u)$, depending on how the function is defined. Although $g_1$ is injective and we assume $Z_1 = \tanh(u)$, without knowing the exact form of $g_1$, we cannot determine whether the underlying variable is $u$ or a reparameterized version $v = h(u)$. Thus, even with an injective proxy function, $u$ remains **unidentifiable up to a monotonic transformation**. To further illustrate the argument that the latent variable $u$ is not identifiable from a single proxy variable—even when the proxy function is injective—we examine the following plot comparison. The solid blue curve shows the original proxy function $g_1(u) = \tanh(u)$, while the dashed orange curve shows the reparameterized function $\tilde{g}_1(v) = \tanh(v - 2)$, where $v = u + 2$.

Although the input variables differ ($u$ vs. $v$), the two curves are identical in shape—they are simply horizontally shifted. Since we only observe the output $Z_1$, and do not know the form of the function or the latent input variable, we cannot determine whether the underlying cause was $u$ or a transformed version $V = h(U)$. This visual comparison confirms that $u$ is **not identifiable** from a single proxy variable, even when the mapping $g_1$ is injective.

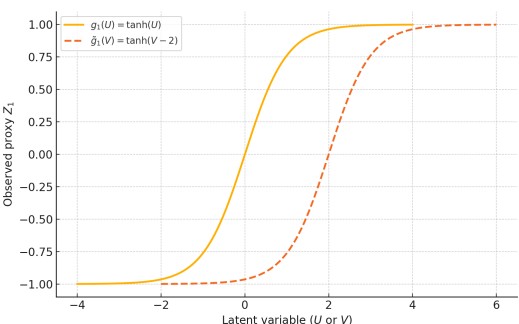

Figure 4: Illustration when k=1.

CASE 2: $k = 2$ — STILL NOT IDENTIFIABLE

Consider the following two proxy variables:

$$Z_1 = g_1(u) = \sin(u), \quad Z_2 = g_2(u) = \cos(u),$$

and define the joint mapping:

$$G(u) = (g_1(u), g_2(u)) = (\sin(u), \cos(u)).$$

This mapping $G(u)$ traces out the unit circle in $\mathbb{R}^2$ as $u$ varies. However, due to the periodic nature of the sine and cosine functions, we have:

$$G(u) = G(u + 2\pi),$$

and more generally:

$$G(u) = G(u + 2n\pi) \quad \text{for any integer } n.$$

This means that all values of $u$ that differ by an integer multiple of $2\pi$ produce the same observed proxy values. As a result, we cannot distinguish between $u$, $u + 2\pi$, $u + 4\pi$, etc., based on $(g_1(u), g_2(u))$ alone. **Therefore, the latent variable $u$ is not identifiable** from these two proxy variables — multiple values of $U$ map to the same point in the observed space. We illustrate the mapping $G(u) = (\sin(u), \cos(u), u)$ by plotting its 3D trajectory as $u$ varies from $0$ to $6\pi$. In this representation, the projection of the curve onto the $(\sin(u), \cos(u))$-plane traces the unit circle repeatedly, while the vertical axis records the increasing values of $u$.

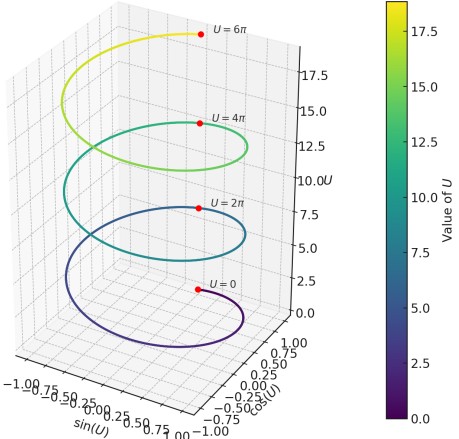

Figure 5: Illustration when k=2.

This visualization clearly demonstrates that $G(u)$ is periodic in its first two components, repeating every $2\pi$. For example, the points $u = 0, 2\pi, 4\pi$, and $6\pi$ all map to the same location in the 2D

plane:$(\sin(u), \cos(u)) = (0, 1)$, but they are separated along the third (vertical) dimension, since: $G(0) = (0, 1)$, $G(2\pi) = (0, 1)$, $G(4\pi) = (0, 1)$, $G(6\pi) = (0, 1)$.

**This confirms that the latent variable $u$ is not identifiable** from the pair of proxy variables $(\sin(u), \cos(u))$, since infinitely many values of $u$ result in the same 2D observation.

CASE 3: $k = 3$ — IDENTIFIABLE UP TO MONOTONIC TRANSFORMATION

Consider the following three proxy variables:

$$Z_1 = g_1(u) = u, \quad Z_2 = g_2(u) = u^2, \quad Z_3 = g_3(u) = u^3, \quad G(u) = (u, u^2, u^3).$$

Then, $G$ is injective and smooth. $G'(u) = (1, 2u, 3u^2) \neq 0$ for all $u \neq 0$. To illustrate injectivity, consider the following examples:

- $u = 1 \Rightarrow G(1) = (1, 1, 1)$
- $u = -1 \Rightarrow G(-1) = (-1, 1, -1)$

Distinct $u$ values yield distinct $G(u)$, so $u$ is identifiable up to a strictly monotonic transformation.

CASE 4: $k > 3$ — IDENTIFIABILITY STILL HOLDS

Consider the following four proxy variables:

$$Z_1 = g_1(u) = u, \quad Z_2 = g_2(u) = u^2, \quad Z_3 = g_3(u) = u^3, \quad Z_4 = g_4(u) = \sin(u).$$

Then, $G(u) = (u, u^2, u^3, \sin(u))$ is injective and smooth. To illustrate injectivity, consider the following examples:

- $u = 1 \Rightarrow G(1) = (1, 1, 1, \sin(1))$
- $u = 2 \Rightarrow G(2) = (2, 4, 8, \sin(2))$,

each $u$ yields a unique $G(u)$, so $u$ is identifiable up to a smooth, strictly monotonic transformation.

# D  A TOY EXAMPLE ILLUSTRATING TENSOR CONSTRUCTION FROM CONDITIONAL MATRICES

In this example, each matrix $M_j \in \mathbb{R}^{r \times 6}$ represents two proxy variables with 3 discrete values each in 6 columns. Specifically:

$$M_1 = \begin{bmatrix} 0.7 & 0.2 & 0.1 & 0.3 & 0.5 & 0.2 \\ 0.4 & 0.4 & 0.2 & 0.6 & 0.3 & 0.1 \\ 0.2 & 0.3 & 0.5 & 0.1 & 0.3 & 0.6 \end{bmatrix},$$

$$M_2 = \begin{bmatrix} 0.5 & 0.3 & 0.2 & 0.6 & 0.3 & 0.1 \\ 0.6 & 0.3 & 0.1 & 0.4 & 0.4 & 0.2 \\ 0.2 & 0.5 & 0.3 & 0.3 & 0.3 & 0.4 \end{bmatrix},$$

$$M_3 = \begin{bmatrix} 0.6 & 0.2 & 0.2 & 0.4 & 0.3 & 0.3 \\ 0.3 & 0.4 & 0.3 & 0.3 & 0.3 & 0.4 \\ 0.5 & 0.3 & 0.2 & 0.2 & 0.3 & 0.5 \end{bmatrix}$$

Each entry represents a conditional probability $P(Z_k = z \mid U = u)$ for the appropriate proxy $Z_k$ value $z$, and latent state $u$. For example, the six entries in the first row of $M_1$ correspond to:

- $P(Z_1 = 0 \mid U = 0)$
- $P(Z_1 = 1 \mid U = 0)$
- $P(Z_1 = 2 \mid U = 0)$
- $P(Z_2 = 0 \mid U = 0)$

- $P(Z_2 = 1 \mid U = 0)$

- $P(Z_2 = 2 \mid U = 0)$

Subsequent rows of $M_1$, as well as all entries in $M_2$ and $M_3$, follow the same pattern for $U = 1$ and $U = 2$, and for proxy variables $Z_3$, $Z_4$, $Z_5$, and $Z_6$. We then construct a tensor $T[i, j, k] \in \mathbb{R}^{6 \times 6 \times 6}$ as:

$$T[i, j, k] = \sum_{r=1}^{3} \pi_r \cdot M_1[r, i] \cdot M_2[r, j] \cdot M_3[r, k], \tag{22}$$

where $\pi_r = P(U = r - 1) = \frac{1}{3}$ is the uniform prior over latent states $U \in \{0, 1, 2\}$. Each index $i, j, k \in \{0, \ldots, 8\}$ corresponds to two proxy values using:

$$(z_a, z_b) = \left( \left\lfloor \frac{\text{index}}{3} \right\rfloor, \text{ index} \bmod 3 \right). \tag{23}$$

For instance, if $i = 1$, $j = 2$, $k = 3$, then:

$$(z_1, z_2) = (0, 1), \quad (z_3, z_4) = (0, 2), \quad (z_5, z_6) = (1, 0).$$

From the matrices above, we have:

| Component | $u = 0$ | $u = 1$ | $u = 2$ |
|---|---|---|---|
| $P(Z_1 = 0 \mid u)$ | 0.7 | 0.4 | 0.2 |
| $P(Z_2 = 1 \mid u)$ | 0.5 | 0.3 | 0.3 |
| $P(Z_3 = 0 \mid u)$ | 0.5 | 0.6 | 0.2 |
| $P(Z_4 = 2 \mid u)$ | 0.1 | 0.2 | 0.4 |
| $P(Z_5 = 1 \mid u)$ | 0.2 | 0.4 | 0.3 |
| $P(Z_6 = 0 \mid u)$ | 0.4 | 0.3 | 0.2 |

We can compute:

$$T[1, 2, 3] = \sum_{u=0}^{2} P(U = u) \cdot P(Z_1 = 0 \mid u) \cdot P(Z_2 = 1 \mid u)$$
$$\cdot P(Z_3 = 0 \mid u) \cdot P(Z_4 = 2 \mid u) \cdot P(Z_5 = 1 \mid u) \cdot P(Z_6 = 0 \mid u).$$

# E  PROOF OF THEOREM 2

**Theorem 2** (Invariance of the ITE under Transformations of the Latent Space)

Let $h : \mathbb{R} \to \mathbb{R}$ be a smooth, strictly monotonic, and invertible function. The individual treatment effect (ITE) is invariant under any smooth, strictly monotonic, and invertible transformation of the latent confounder. That is, if $\hat{U} = h(U)$, then $\text{ITE}(\hat{U}_i) = \text{ITE}(U_i)$.

*Proof.* Let $\hat{U}_i = h(U_i)$, where $h$ is a smooth, strictly monotonic, and invertible transformation. Since $h$ is invertible, we have $U_i = h^{-1}(\hat{U}_i)$. From the structural equation defined in equation 1, the counterfactual outcome depends on $T$ and $U$, and not on the specific representation of $U$. Thus, for any treatment $t \in \{0, 1\}$, we have:

$$\mathbb{E}[Y_i \mid \text{do}(T = t), \hat{U}_i] = \mathbb{E}[Y_i \mid \text{do}(T = t), h^{-1}(\hat{U}_i)] = \mathbb{E}[Y_i \mid \text{do}(T = t), U_i]. \tag{24}$$

Therefore, we can conclude:

$$\begin{aligned} \text{ITE}(\hat{U}_i) &= \mathbb{E}[Y_i \mid \text{do}(T = 1), \hat{U}_i] - \mathbb{E}[Y_i \mid \text{do}(T = 0), \hat{U}_i] \\ &= \mathbb{E}[Y_i \mid \text{do}(T = 1), U_i] - \mathbb{E}[Y_i \mid \text{do}(T = 0), U_i] = \text{ITE}(U_i). \end{aligned} \tag{25}$$

$\square$

# F    PROOF OF PROPOSITION 1

> **Proposition 1** (Benefits of GP Priors for Latent Confounder Regularization in Causal Models with Unobserved Confounding)
>
> Let $U = (u_1, \ldots, u_n)^\top$ denote the latent confounder values corresponding to $n$ observed samples. Suppose each sample is associated with observed data $Z^i$, where $Z^i \in \mathbb{R}$ represents noisy proxies of $U$, via the proxy structural equation denoted by equation 2. Let the latent confounders be endowed with a GP prior:
>
> $$U \sim \mathcal{GP}(0, K(Z^i, Z^j)),$$
>
> where $K$ is a positive-definite kernel function applied to two observed variables $Z^i$, $Z^j$. Then:
>
> 1.  The GP prior does not violate identifiability of the causal effect (e.g., ITE).
>
> 2.  For any smooth and invertible transformation $h : \mathbb{R} \to \mathbb{R}$, defining $\tilde{U} = h(U)$, the GP prior still encodes the same relative geometry via an induced kernel.

*Proof.* From Theorem 2, the ITE remains invariant under any smooth, strictly monotonic, and invertible transformation $h$ of the latent confounder $U$, i.e., $\mathrm{ITE}(h(U)) = \mathrm{ITE}(U)$. The GP prior over $U \sim \mathcal{GP}(0, k(Z^i, Z^j))$ encourages smoothness by enforcing that similar inputs $Z^i \approx Z^j$ induce similar latent values $u^i \approx u^j$, and regularizes only the *relative geometry* of the latent space, without constraining its absolute coordinate values. Hence the GP prior does not conflict with identifiability and preserves valid estimation of causal effects. For any monotonic bijection $h$, we may write $\tilde{u}_i = h(u_i)$. Under this transformation, the learned decoder $g$ or treatment/outcome functions $f$ can be reparameterized accordingly (e.g., $\tilde{g} = g \circ h^{-1}$). The GP kernel matrix $\tilde{K}_{ij} = k(h^{-1}(Z^i), h^{-1}(Z^j))$ induces a valid alternative prior with identical relative geometry. Therefore, the learned model remains observationally and causally equivalent. $\square$

