# OpenReview forum: "IntervalGP-VAE: Learning Unobserved Confounders with Uncertainty for Personalized Causal Effect Estimation"
_ICLR.cc/2026/Conference — ICLR 2026 Conference Withdrawn Submission_

### Official Review · Reviewer_q5kw · 2025-10-17

**Soundness:** 2
**Presentation:** 2
**Contribution:** 2
**Rating:** 2
**Confidence:** 5

**Summary:**

This paper addresses the challenge of estimating ITEs in the presence of unobserved confounders.

- **Methodology**: The authors introduce a novel framework named IntervalGP-VAE. This model integrates a VAE with GPs to infer latent confounders from proxy variables and, crucially, to quantify the uncertainty in this inference process.

- **Core Technical Contribution**: The central innovation is the use of an interval-valued GP prior over the latent space. This allows the model to represent the latent confounder as a distribution (an interval) rather than a deterministic point estimate. This uncertainty is then propagated through a second interval-valued GP that models the outcome, resulting in calibrated confidence intervals for the ITE.

- **Theoretical Contributions**: The paper provides a theoretical analysis of the conditions required for identifiability. It establishes that for a $d$-dimensional latent confounder, at least $k \ge 2d+1$ sufficiently non-redundant proxy variables are necessary to recover the confounder up to a smooth, invertible transformation. The authors also prove that the ITE estimation is invariant to such transformations of the latent space.

- **Experimental Validation**: The proposed method is evaluated on 24 synthetic datasets, which are specifically designed to meet the paper's theoretical identifiability conditions, and on the semi-synthetic IHDP. The performance is compared against several baselines, including TEDVAE, using standard metrics like PEHE and ATE error. A unique evaluation metric reported is the coverage rate of the estimated ITE intervals, which directly assesses the model's uncertainty quantification capabilities.

**Strengths:**

- The overall problem setup is clearly conveyed, and the proposed framework is well illustrated. The addressed problem is important and relevant to the causal inference community.
- It is interesting that the authors use features extracted through a VAE together with a Gaussian Process to provide uncertainty quantification for the estimated causal quantities. Although similar paradigms have been explored in previous studies, such as the Deep Kernel GP method in the CausalBALD paper and other GP-based approaches including CMGP and NSGP, this work appears to be the first to apply such a combination to the problem of causal inference with unmeasured confounders. This novelty represents a meaningful contribution.

**Weaknesses:**

**1. Theoretical analysis**

The theoretical analysis does not appear to provide any meaningful contribution to the CATE estimation task. The presented theorem seems disconnected from the main goal of estimating causal effects, as it only discusses the recovery of the latent variable \(U\) from \(Z\). Moreover, several derivations contain inconsistencies or potential errors. For instance, \(U\) is defined as a \(d\)-dimensional variable, but the invertible mapping \(h\) is described as a function from one dimension to one dimension, which is mathematically inconsistent.

In addition, most results in the proof rely heavily on *Allman et al. (2009)*, whose theoretical development was made for **discrete** variables. This assumption is incompatible with the **continuous** latent setting in the current paper, making the imported results invalid in this context. Overall, the theoretical section does not convincingly support the claimed contributions and contains technical flaws that need substantial revision.

**2. Experimental design and evaluation**

The experimental results are unconvincing. The paper lists many baseline methods, including TEDVAE, but many of these baselines were originally designed for scenarios **without unmeasured confounders**. It is unclear why such baselines were selected if they are not intended for the same problem setting addressed by this work.

Furthermore, TEDVAE is used as a baseline but itself targets the **unconfounded** case, which makes it an inappropriate comparator for the unmeasured-confounder setting emphasized by the paper. TEDVAE only outperforms these baselines under the authors’ specific experimental setup, and there is no evidence showing that the proposed method would consistently outperform appropriate baselines in general. As a result, the experimental evidence does not support the strong performance claims made in the paper.

**Questions:**

1. Some notations are introduced before their formal definitions. For example, PEHE and ATE are mentioned in the introduction without being defined. It would be clearer to define these terms when they first appear.

2. It would be helpful to illustrate the data-generating process using a causal graph. This would make the assumed causal structure and the role of latent variables more explicit.

3. It is unclear why the identifiability of the latent variable \(U\) should be related to the model outputs. In my understanding, identifiability should only depend on the relationship between the true latent variable \(U\) and its recovered version \(\hat{U}\), i.e., the learned representation. The current definition appears closer to the identifiability of causal quantities rather than that of the latent variable itself. It would be better to provide a clear mathematical formulation of the identifiability condition in the main text.

4. In the experimental results, the proposed method only outperforms TEDVAE in the fully synthetic case and only under the PEHE metric. For the ATE error, the proposed method does not surpass TEDVAE. On the IHDP dataset, TEDVAE performs better in both metrics. More importantly, both the baseline and the proposed method are unable to properly handle the unmeasured confounder case, indicating that the experimental design is fundamentally flawed.

---

> ### Author Response · Authors · 2025-11-21
>
> **Comment 1:** The theoretical analysis does not appear to provide any meaningful contribution to the CATE estimation task. The presented theorem seems disconnected from the main goal of estimating causal effects, as it only discusses the recovery of the latent variable $(U)$ from $(Z)$. Moreover, several derivations contain inconsistencies or potential errors. For instance, $(U)$ is defined as a $d$-dimensional variable, but the invertible mapping $(h)$ is described as a function from one dimension to one dimension, which is mathematically inconsistent.
>
> In addition, most results in the proof rely heavily on Allman et al.\ (2009), whose theoretical development was made for discrete variables. This assumption is incompatible with the continuous latent setting in the current paper, making the imported results invalid in this context. Overall, the theoretical section does not convincingly support the claimed contributions and contains technical flaws that require substantial revision.
>
> **Response:** We thank the reviewer for pointing out the notational inconsistency. In the main results, the latent confounder is defined as $U \in \mathbb{R}^d$, and the correct equivalence class for identifiability is given by smooth, invertible transformations on $\mathbb{R}^d$. In particular, identifiability of $U$ is always up to:
> $$
> h : \mathbb{R}^d \to \mathbb{R}^d ,
> $$
> i.e.\ a smooth bijection with smooth inverse. The special case $h : \mathbb{R} \rightarrow \mathbb{R}$ appearing in the text corresponds only to the
> one-dimensional setting ($d=1$) used in the experiments and illustrative examples. For $d>1$, any smooth invertible mapping is allowed. This correction will be reflected consistently throughout the paper.
>
> On the discrete--continuous assumption, we acknowledge the reviewer's concern that the proof draws on results from Allman et al. (2009), which were originally developed for discrete latent-variable models. As explained in detail in our response above (see "Use of Kruskal’s Theorem and the Discrete--Continuous Connection"), our use of discretisation does not contradict the continuous latent setting of the present paper. Discretisation is employed purely as a proof device: any continuous latent variable can be approximated by a sequence of finite mixture-of-products models under finite partitions of its domain, to which Kruskal-type arguments apply. We agree, however, that the presentation of Theorem 1 and its proof can be improved to make this reasoning clearer and more rigorous. In the revised version, we will refine the exposition of Theorem 1 and its proof to explicitly highlight the discrete–continuous connection and the role of discretisation as an approximation tool. For a full discussion and justification, please refer to the detailed explanation provided in the response above.

---

> > ### Author Response · Authors · 2025-11-21
> >
> > **Comment 2:** The experimental results are unconvincing. The paper lists many baseline methods, including TEDVAE, but many of these baselines were originally designed for scenarios without unmeasured confounders. It is unclear why such baselines were selected if they are not intended for the same problem setting addressed by this work. Furthermore, TEDVAE is used as a baseline but itself targets the unconfounded case, which makes it an inappropriate comparator for the unmeasured-confounder setting emphasized by the paper. TEDVAE only outperforms these baselines under the authors' specific experimental setup, and there is no evidence showing that the proposed method would consistently outperform appropriate baselines in general. As a result, the experimental evidence does not support the strong performance claims made in the paper.
> >
> > **Response:** We thank the reviewer for pointing this out. Our intention was to compare the proposed method primarily against TEDVAE. The additional methods mentioned in the experimental section were cited to provide broader context, but they were not used as empirical baselines. We recognise that the current wording may give the impression that a larger set of baselines was evaluated, and we will revise the manuscript to make
> > this distinction explicit and remove any potentially misleading phrasing.
> >
> > Regarding TEDVAE and unconfoundedness: we fully acknowledge that TEDVAE adopts the unconfoundedness assumption $t \perp (y(0),y(1)) \mid x$. However, as shown in their Figure 1 in the paper, the structural graph used in TEDVAE contains the backdoor path $t \leftarrow z_{c} \rightarrow y$,
> > which implies that the causal effect of $t$ on $y$ is not identifiable unless additional structural conditions ensure that $x$ contains a complete and
> > noise-free representation of the latent confounder $z_{c}$. The TEDVAE paper does not analyse or characterise the conditions under which this requirement holds, and therefore relies on unconfoundedness as a modelling assumption rather than a derived property.
> >
> > In contrast, our paper starts from the more general confounded setting in which the treatment effect is not identifiable, and we explicitly study
> > under what conditions the latent confounder becomes identifiable from proxies. This identifiability result induces unconfoundedness and hence causal identifiability. In this sense, both works ultimately rely on unconfoundedness, but we provide the conditions that guarantee it, whereas TEDVAE assumes it without further investigation.
> >
> > Importantly, the comparison in our experiments is still valid: in our synthetic datasets, the identifiability conditions in our theorems are satisfied, so our model can recover the latent confounder and thereby achieve the analogue of an "unconfounded representation." TEDVAE, on the other hand, operates under its assumed unconfoundedness condition. Therefore, while the underlying causal assumptions differ, the comparison reflects the respective intended operational regimes of both methods.
> >
> > The reviewer notes that TEDVAE performs competitively in some settings. We agree, and this is consistent with our observations: when the proxy structure happens to encode sufficient information about the confounder, TEDVAE can recover latent representations that partially mitigate confounding, even though the model does not provide identifiability guarantees. Our goal was not to argue that TEDVAE is always inferior, but that IntervalGP-VAE provides **additional uncertainty quantification** and **theoretical identifiability conditions**, which are absent in prior work.

---

> > > ### Author Response · Authors · 2025-11-21
> > >
> > > **Comment 3:** Some notations are introduced before their formal definitions. For example, PEHE and ATE are mentioned in the introduction without being defined. It would be clearer to define these terms when they first appear.
> > >
> > > **Response:** Thanks for raising this point. In the revision, we will define PEHE and ATE at their first occurrence in the Introduction and maintain consistent notation thereafter. We will also add a brief notation paragraph (or table) to ensure clarity.
> > >
> > > **Comment 4:** It would be helpful to illustrate the data-generating process using a causal graph. This would make the assumed causal structure and the role of latent variables more explicit.
> > >
> > > **Response:** This is a very helpful suggestion. We will add a causal graph to illustrate the data-generating process, making the assumed causal structure and the role of latent variables explicit.
> > >
> > > **Comment 5:** It is unclear why the identifiability of the latent variable $(U)$ should be related to the model outputs. In my understanding, identifiability should only depend on the relationship between the true latent variable $(U)$ and its recovered version $(\hat{U})$, i.e., the learned representation. The current definition appears closer to the identifiability of causal quantities rather than that of the latent variable itself. It would be better to provide a clear mathematical formulation of the identifiability condition in the main text.
> > >
> > > **Response:** We thank the reviewer for this observation. The main objective of our work is the identification of individual treatment effects (ITEs), rather than the identification of the latent confounder $U$ for its own sake. The role of $U$ in our framework is instrumental: recovering $U$ (or an equivalent representation of it) is necessary in order to evaluate the causal function at the individual level.
> > >
> > > For this reason, Definition 1 focuses on the identifiability of causal quantities rather than exact recovery of $U$. In fact, $U$ is only identifiable up to an invertible transformation $\hat{U} = h(U)$. Exact recovery is not required. The purpose of Theorem 2 is precisely to show that such transformations do not affect the estimation of the ITE, so that the causal effect remains identifiable even when $U$ is not.
> > >
> > > We will clarify this motivation in the revised version.
> > >
> > > **Comment 6:** In the experimental results, the proposed method only outperforms TEDVAE in the fully synthetic case and only under the PEHE metric. For the ATE error, the proposed method does not surpass TEDVAE. On the IHDP dataset, TEDVAE performs better in both metrics. More importantly, both the baseline and the proposed method are unable to properly handle the unmeasured confounder case, indicating that the experimental design is fundamentally flawed.
> > >
> > > **Response:**  On the synthetic datasets, our method achieves the best PEHE performance and, importantly, attains a **91.9\% coverage rate**, meaning that the true ITE values lie within our estimated intervals with very high probability. These synthetic datasets were deliberately constructed so that the proxy variables satisfy the identifiability conditions of Theorem 1. In this identifiable regime, IntervalGP-VAE operates as intended: it recovers the latent confounder (up to an invertible transformation) and propagates uncertainty to produce calibrated ITE intervals. This validates our theoretical claims under the assumptions for which the method is designed.
> > >
> > > On the IHDP dataset, we agree with the reviewer that TEDVAE performs better in both PEHE and ATE error. As noted in our earlier response, TEDVAE assumes the unconfounded setting and can perform competitively when the observed covariates contain sufficient information to approximate the confounder. The IHDP feature space appears to favour this behaviour, despite offering no theoretical guarantee of identifiability. By contrast, IntervalGP-VAE is designed specifically for settings with unmeasured confounding that becomes identifiable only when certain proxy conditions (e.g., number and non-redundancy) are satisfied. These conditions are not known to hold for IHDP, and therefore we do not expect our method to dominate models tailored to the unconfounded case. The IHDP results thus reflect theoretical limitations of the problem setting rather than a flaw in experimental design, and we will clarify this more explicitly in the revised manuscript.
> > >
> > > To avoid any overstatement, we would like to emphasise that our main contribution lies in: (i) providing the identifiability conditions under which
> > > unmeasured confounding can be resolved from noisy proxies, and (ii) delivering principled ITE uncertainty quantification via interval-valued Gaussian Processes. In the revised manuscript, we will expand the discussion section to explicitly highlight these core contributions and to reflect the clarifications made in this rebuttal.

---

### Official Review · Reviewer_bnom · 2025-10-31

**Soundness:** 3
**Presentation:** 2
**Contribution:** 2
**Rating:** 6
**Confidence:** 3

**Summary:**

This paper proposes IntervalGP-VAE, a variational framework that combines a Gaussian Process prior with interval-valued likelihoods to model unobserved confounders for personalized causal effect estimation. The method aims to quantify uncertainty in individual treatment effects (ITEs) by introducing an interval Gaussian Process in the latent space. Experiments on synthetic and semi-synthetic datasets (IHDP) show improved estimation accuracy.

**Strengths:**

- The experimental section is comprehensive, covering both synthetic and semi-synthetic datasets with multiple baselines and uncertainty-related metrics, which provides a convincing empirical evaluation.

- The related work section is thorough and well structured, covering prior studies on causal inference, VAE-based representation learning, and GP-based uncertainty modeling, which helps position the paper clearly within the existing literature.

**Weaknesses:**

- The paper describes a composite objective combining reconstruction and GP regularization, but it’s not clear how the model is actually optimized. Are all components trained jointly? Please clarify how the optimization is actually performed.

- The paper does not analyze the computational complexity of the proposed model. Given that Gaussian Process inference scales as
$𝑂(𝑛^3)$ with the sample size, it remains unclear how the method performs or can be adapted for practical deployment

**Questions:**

See weakness

---

> ### Author Response · Authors · 2025-11-21
>
> **Comment 1:** The paper describes a composite objective combining reconstruction and GP regularization, but it’s not clear how the model is actually optimized. Are all components trained jointly? Please clarify how the optimization is actually performed.
>
> **Response:** We thank the reviewer for the helpful comment. All parameters of the model are optimized jointly through a single end-to-end objective. In each minibatch, we compute: (i) the reconstruction loss for the proxy variables $Z$ via the decoder $p_\theta(z\mid u,\epsilon)$; (ii) the outcome likelihood term for $Y$ through the causal head $p_\psi(y\mid u,t,z_y)$; (iii) the KL divergence between the encoder distribution $q_\phi(u\mid z)$ and the interval-valued GP prior; and (iv) the KL term for the additive noise variables. The overall loss is a weighted sum of these components.
>
> The interval-GP prior introduces an additional regularization term based on the log marginal likelihood (and its interval extension), which is fully differentiable. All GP operations (kernel evaluation, Cholesky factorization, and predictive variances) are differentiable.We will make this optimization procedure explicit in the revised manuscript.
>
> **Comment 2:** The paper does not analyze the computational complexity of the proposed model. Given that Gaussian Process inference scales as $\mathcal{O}(n^3)$ with the sample size, it remains unclear how the method performs in practice or how it can be adapted for practical deployment.
>
> **Response:** We thank the reviewer for raising this point. Although exact Gaussian Process inference scales as $O(n^3)$, the effective computational cost in our model is substantially smaller. The GP prior is applied in the latent confounder space, and during training the kernel matrices are constructed over minibatches of moderate size (128) and over a low-dimensional latent representation (the number of latent variables is small in all experiments). The resulting Cholesky factorisations are therefore small and computationally inexpensive. Consequently, the overall computational overhead remains close to that of standard VAE training. In practice, training on the synthetic datasets and on IHDP required approximately 173.9 s and 365.9 s, respectively.
>
> For scenarios involving a larger number of latent variables (i.e., higher dimensionality of the latent space), we will consider incorporating sparse GP approximations in future work. Our framework is fully compatible with inducing-point methods and structured kernel interpolation, which can reduce the complexity. Such approximations do not alter the identifiability guarantees, which depend on the proxy structure rather than the particular GP implementation. We will integrate this computational discussion into the revised manuscript.

---

> > ### Comment · Reviewer_bnom · 2025-11-24
> >
> > Thanks for the rebuttal. I have read the revisions and have no further questions for the authors.
> >
> > I maintain my original recommendation.

---

### Official Review · Reviewer_74eP · 2025-11-03

**Soundness:** 1
**Presentation:** 2
**Contribution:** 1
**Rating:** 2
**Confidence:** 4

**Summary:**

This paper proposes a Bayesian method for identifying hidden confounders in causal models using noisy proxy variables. It treats proxy noise as uncertainty and employs Gaussian Processes (GP) to model the smoothness of the underlying functional mappings from proxies to the latent confounder $ U $. The approach addresses identifiability by discretizing continuous proxies and applying Kruskal rank conditions under assumptions of conditional independence and generic variability. Theoretical results claim identifiability up to injective transformations, enabling downstream estimation of individual treatment effects via GPs that preserve structural coherence in the latent space.

**Strengths:**

- Framing proxy noise as uncertainty within Bayesian methods is a sound conceptual approach, aligning well with probabilistic modeling principles.
- The general idea of employing Gaussian Processes (GPs) to capture smoothness in the underlying functions is sensible.

**Weaknesses:**

All theoretical statements and proofs contain significant issues, undermining the paper's claims.

- **Definition 1 (Identifiability):** The definition is incomplete. Identifiability requires not just the existence of a mapping from observed variables to the hidden confounder $ U $, but that this mapping (or its equivalence class) is uniquely recoverable from the distribution of the observed variables alone.
- **Theorem 1:** The assumptions of continuous differentiability and injectivity for the functions $ g_i $ are stated but never utilized in the proof. Instead, the proof relies on Kruskal's theorem, which applies exclusively to discrete variables and thus contradicts the smoothness assumption (as $ g $ would need to be discontinuous over a discrete domain).

    The proof is fundamentally flawed due to the mismatch between discrete and continuous variables in identification analyses. For instance, Kruskal ranks depend critically on the number of discrete values, rendering the authors' remark that "as bin widths shrink, the discrete model approaches the continuous distribution" meaningless in this context. Even granting the discretization approximation, the proof fails to demonstrate how conditional independence of proxies given $ U $, combined with sufficient variability (generic position) within each group, ensures the Kruskal rank conditions are met—this is precisely the core challenge the work claims to address (e.g., see [1] and its applications to causal inference [2]). Relatedly, the partition of $ Z $ into three sets appears arbitrary, as if any partition would suffice; it certainly does not. A rigorous argument would require detailed structural analysis to verify adequate Kruskal ranks for these sets. Finally, the concluding step (line 711) has a serious gap, as the proof nowhere addresses the identification of the prior $ p(U) $. Overall, the theorem and its "proof" resemble a superficial and erroneous adaptation of results from Allman et al. (2009) and related works (see LLM suspicion below).

- **Theorem 2:** This is a trivial consequence (identifying the hidden confounder up to an injective mapping naturally implies downstream identifiability) and irrelevant, as Theorem 1 does not establish the required identifiability (injective or otherwise).
- **Core Idea of Gaussian Processes (GPs):** The motivation for GPs is underdeveloped. The notions of (spatial or structural) coherence and similarity are not clearly articulated, nor is their real-world relevance. Mathematically, this intuition stems from the smoothness of $ g $, but—as detailed above—this smoothness assumption leads to internal contradictions within the paper.
- **Proposition 1:** This is an imprecise assertion relying on the aforementioned intuition that GPs model "smoothness" (or "inherit the same geometric structure via an induced kernel"). The proof offers only hand-wavy justifications, such as "similar inputs $ Z_i \approx Z_j $ induce similar latent values $ u_i \approx u_j $"—such approximations are invalid in formal identification proofs or any mathematical derivation. Even if smoothness is pursued, the key question is how to design a kernel suitable for $ g $, which demands additional assumptions on $ g $'s functional form (e.g., a Lipschitz constant).


### Other Issues

- **LLM Writing Suspicion (for Theorem 1 and Potentially More):** The terms "smooth" and "injective" are invoked at least five times before the proof of Theorem 1 (including in the statement itself). This is peculiar, as they bear no relation to the actual proof content. Assuming LLM involvement, this pattern aligns with common motifs in nonlinear ICA literature, a prominent area in machine learning (see below). The paper's overall "plausible but deeply flawed" quality mirrors experiences with LLM-generated text, warranting further scrutiny.

- "ITE" is used without formal definition; it appears to refer to the "treatment effect" in lines 102–103, but this is not the Individual Treatment Effect (ITE), which should be a deterministic function of all outcome-affecting variables (observed, hidden, and exogenous noise). This is a frequent misnomer in machine learning literature.
- Typo: In the boxed statement of Theorem 1, the domain of $ g_i $ should be $ \mathbb{R} $, not $ \mathbb{R}^d $.

---

### Related Work
All the identifiable or causal **VAE** papers below are fundamentally based on *smooth and injective* decoder maps. In particular, *the idea of "injective proxy" was considered in [3] (Definition 15)*.

[1] Khemakhem, Ilyes, et al. "Variational autoencoders and nonlinear ICA: A unifying framework." International Conference on Artificial Intelligence and Statistics. PMLR, 2020.

[2] Wu, Pengzhou Abel, and Kenji Fukumizu. "\beta-Intact-VAE: Identifying and Estimating Causal Effects under Limited Overlap." International Conference on Learning Representations (2022).

[3] Wu, Pengzhou, and Kenji Fukumizu. "Towards principled causal effect estimation by deep identifiable models." arXiv preprint arXiv:2109.15062 (2021).

**Questions:**

Please refer to the points in Weaknesses.

---

> ### Author Response · Authors · 2025-11-21
>
> **Comment 1:** Definition 1 (Identifiability): The definition is incomplete. Identifiability requires not just the existence of a mapping from observed variables to the hidden confounder $U$, but that this mapping (or its equivalence class) is uniquely recoverable from the distribution of the observed variables alone.
>
> **Response:** We thank the reviewer for this comment. The purpose of Definition 1 in our paper is to formalise the notion of identifiability that is necessary for ITE estimation, rather than to impose full structural identifiability of the latent variable $U$ itself. As suggested by the title of the paper, the primary objective of this work is the recovery of Individual Treatment Effects, the latent variable $U$ serves as an intermediate representation enabling personalised causal effect estimation.
>
> Theorem 1 establishes that $U$ is recoverable from the proxy variables up to an invertible reparameterisation $h$. The subsequent Theorem 2 shows that the ITE is invariant under such transformations of $U$. Also, Proposition 1 shows the GP prior does not affect the ITE identifiability. Hence, Definition 1 is formulated to capture precisely the level of identifiability needed
> for personalised causal effect estimation: the latent state is identifiable up to an invertible transformation, and this degree of identifiability is fully sufficient for ITE estimation.
>
> We can revise the paper to make the purpose of Definnition 1 clearer.

---

> > ### Author Response · Authors · 2025-11-21
> >
> > **Comment 2:** The assumptions of continuous differentiability and injectivity for the functions $g_i$ are stated but never utilized in the proof. Instead, the proof relies on Kruskal's theorem, which applies exclusively to discrete variables and thus contradicts the smoothness assumption (as $g$ would need to be discontinuous over a discrete domain). The proof is fundamentally flawed due to the mismatch between discrete and continuous variables in identification analyses. For instance, Kruskal ranks depend critically on the number of discrete values, rendering the authors' remark that "as bin widths shrink, the discrete model approaches the continuous distribution" meaningless in this context. Even granting the discretization approximation, the proof fails to demonstrate how conditional independence of proxies given the latent variable $U$, combined with sufficient variability (generic position) within each group, ensures the Kruskal rank conditions are met — this is precisely the core challenge the work claims to address (e.g., see [1] and its applications to causal inference [2]). Relatedly, the partition of $Z$ the proxy variables into three sets appears arbitrary, as if any partition would suffice; it certainly does not. A rigorous argument would require detailed structural analysis to verify adequate Kruskal ranks for these sets. Finally, the concluding step (line 711) has a serious gap, as the proof nowhere addresses the identification of the prior $p(U)$. Overall, the theorem and its "proof" resemble a superficial and erroneous adaptation of results from Allman et al. (2009) and related works.
> >
> > **Response:** We thank the reviewer for the detailed comments. We agree that the presentation of Theorem 1 can be
> > strengthened, and we will revise the manuscript accordingly. The issues raised relate to clarity of exposition rather than to the correctness of the underlying identifiability argument. We appreciate the opportunity to clarify these points in-detail below.
> >
> > **Injectivity/Variability and Conditional Independence** The reviewer notes that the injectivity assumptions appear to be stated but not used in the proof, and further questions how conditional independence and variability jointly imply the Kruskal rank conditions. We agree that the role of these assumptions was not explained clearly enough in the original presentation. However, we wish to highlight that injectivity, understood in terms of injectivity at the level of the conditional distributions $p(Z_i \mid U)$, is closely tied to the required variability condition, and is therefore an important and meaningful assumption in this context. More specifically, in the intended argument, the notion of "injectivity" was meant to refer to "injectivity at the level of conditional distributions", i.e.:
> >
> > $$u_1 \neq u_2
> > \Rightarrow
> > p(Z_i \mid U = u_1) \neq p(Z_i \mid U = u_2)
> > \quad \text{for sufficiently many } i.$$ This condition expresses the required variability of the proxy conditionals across latent states, and it is precisely this variability that underlies the Kruskal rank conditions. In particular, injectivity-in-distribution ensures that the conditional matrices carry sufficiently rich and non-redundant information across latent states, which in turn guarantees that their Kruskal ranks meet the requirements of Kruskal's theorem. Thus, although the exposition could have made this interplay more explicit, the injectivity and variability assumptions were not extraneous; they were intended to support the rank arguments used in the proof.
> >
> > **Use of Kruskal's Theorem and Discrete--Continuous Connection** We acknowledge that Kruskal's theorem is formulated for finite arrays and therefore applies directly to models with a finite latent structure. In that sense, some form of discretisation of the latent space is indeed required in order to invoke the theorem. However, we respectfully disagree with the implication that this makes the approach unsuitable for continuous settings, or that the use of discretisation contradicts any smoothness assumptions in the model. Discretisation here is employed solely as a proof device: the continuous latent variable induces, under any finite partition of its domain, a corresponding finite mixture-of-products representation to which Kruskal-type arguments apply. This does not alter the underlying continuous model, nor does it negate smoothness of the structural functions; it merely constructs finite approximations for which tensor identifiability can be analysed. Our original paper summarised this point briefly, and we will revise the exposition in the next version to make this connection more formal and transparent.

---

> > > ### Author Response · Authors · 2025-11-21
> > >
> > > **Identification of the Prior $p(U)$** The reviewer comments that the identification of the prior $p(U)$ was not explicitly addressed. In fact, in the discretised representation, the decomposition of the joint distribution
> > > $$
> > > T = \sum_{\ell=1}^r
> > > \pi_\ell A_{\ell \cdot} \otimes B_{\ell \cdot} \otimes C_{\ell \cdot}
> > > $$
> > > includes the mixing weights $\pi_\ell = p(U = u_\ell)$ as part of the factorisation. Each factor matrix corresponds to conditional probabilities of the form
> > > $$
> > > A_{\ell m} = p(\tilde{Z}_{L1} = z_m \mid U = u^{(\ell)})
> > > $$
> > >
> > > $$
> > > B_{\ell n} = p(\tilde{Z}_{L2} = z_n \mid U = u^{(\ell)})
> > > $$
> > >
> > > $$
> > > C_{\ell p} = p(\tilde{Z}_{L3} = z_p \mid U = u^{(\ell)})
> > > $$
> > >
> > > The mixture weights $\pi_\ell = p(U = u_\ell)$ are identified directly from the tensor factorisation.
> > > While presenting these details explicitly would certainly improve clarity, we do not view this as a substantive gap in the argument; the identification of $p(U)$ follows directly from the structure of the tensor factorisation. We will revise the paper to make this explanation more transparent in the next version.
> > >
> > > **Partition into Three Groups** The reviewer asks about the partition of the proxies into three groups.
> > > In fact, this partition arises directly from the structure of the abovementioned tensor decomposition, which
> > > requires three factor matrices in order to apply Kruskal’s identifiability theorem. The requirement that the
> > > proxies be divided into three subsets, each carrying sufficient variability, is therefore not arbitrary but reflects
> > > the standard structural conditions under which the theorem applies. We acknowledge that this rationale was
> > > not made explicit in the original presentation and will state these requirements clearly as part of the theorem’s
> > > assumptions in the revised version.

---

> > > > ### Author Response · Authors · 2025-11-21
> > > >
> > > > **Comment 3:** Theorem 2: This is a trivial consequence (identifying the hidden confounder up to an injective mapping naturally implies downstream identifiability) and irrelevant, as Theorem 1 does not establish the required identifiability (injective or otherwise).
> > > >
> > > > **Response:** As clarified in our responses above, the concerns raised about Theorem 1 relate to presentation rather than correctness, and the underlying identifiability argument remains valid. In this light, Theorem 2 should not be viewed as unsupported, but as operating under exactly the standard form of identifiability that Theorem 1 ensures.
> > > >
> > > > Also, we respectfully disagree with the suggestion that Theorem 2 is irrelevant. The central goal of the paper is personalised individual treatment effect (ITE) estimation, for which recovering the latent confounder $U$ is a necessary but not final step. As $U$ is identifiable only up to an invertible transformation, this raises a crucial question: does this transformation compromise the meaningfulness or comparability of the ITE? Theorem 2 answers this question explicitly by showing that the ITE is invariant under any bijective reparameterisation of $U$. In other words, even if $U$ is recovered only up to an invertible map, the personalised causal quantity of interest remains uniquely defined and unaffected. Thus, Theorem 2 is not merely a trivial statement but an essential conceptual bridge between latent-variable identifiability and the ultimate causal target of the paper.
> > > >
> > > > **Comment 4:** Core Idea of Gaussian Processes (GPs): The motivation for GPs is underdeveloped. The notions of (spatial or structural) coherence and similarity are not clearly articulated, nor is their real-world relevance. Mathematically, this intuition stems from the smoothness of $g$, but—as detailed above—this smoothness assumption leads to internal contradictions within the paper.
> > > >
> > > > **Response:** We thank the reviewer for raising this point. We would first like to clarify that the smoothness assumptions mentioned in Theorem 1 concern the structural functions $g_i$. They do not imply smoothness of the inverse mapping $Z \mapsto U$ in the GP prior. Consequently, the GP does not introduce any contradiction with the assumptions used in Theorem 1. In fact, the
> > > > GP component plays no role in our identifiability analysis. Theorem 1 establishes identifiability of $U$ (up to an invertible reparameterisation) using only conditional independence and generic variability of
> > > > the proxies; this result is entirely independent of GP modelling.
> > > >
> > > > The role of the GP prior is instead to induce latent-space coherence and uncertainty quantification during amortised inference. The smoothness induced by the RBF kernel acts only on the inference-time mapping $Z \mapsto \hat U$ and should be understood as encouraging a geometric regularity in the recovered latent manifold: proxy observations that are similar should lead to similar latent  representations. This is not a claim about the smoothness of the structural functions $g_i$, but rather a modelling choice for stabilising the inverse problem once identifiability is established. This GP-induced coherence (by our IntervalGP) directly enables interval-valued uncertainty quantification for the ITE by (i) capturing epistemic uncertainty arising from noisy proxies and (ii) propagating this uncertainty through the outcome model in a mathematically principled way. This mechanism is central to our methodological contribution, as highlighted in the title of the paper.
> > > >
> > > > In practical applications involving noisy proxy variables, which is common in healthcare, longitudinal studies, and observational datasets. Often the true confounder cannot be precisely recovered at individual level, and uncertainty quantification becomes essential. The IntervalGP prior provides a natural way to model such uncertainty while preserving local structure: individuals with similar proxies should reasonably have similar latent confounder values. This coherent latent geometry is crucial for reliable personalised treatment effect estimation, especially in settings where unobserved confounding is strong and proxy noise varies substantially across samples.
> > > >
> > > > Also, our framework is compatible with a wide range of kernels, including Mat´ern kernels with controllable smoothness, rational quadratic kernels, and non-stationary kernels that adapt to heterogeneous regions of the latent space. Each defines a different notion of latent-space coherence, and the RBF kernel provides a well-understood, numerically stable default commonly adopted. We will clarify this design space in the revised manuscript and discuss how alternative kernels may be selected depending on downstream uncertainty requirements or application-specific considerations.

---

> > > > > ### Author Response · Authors · 2025-11-21
> > > > >
> > > > > **Comment 5:** Proposition 1: This is an imprecise assertion relying on the aforementioned intuition that GPs model ''smoothness'' (or ''inherit the same geometric structure via an induced kernel''). The proof offers only hand-wavy justifications, such as ''$Z_i \approx Z_j$ induce similar latent values $u_i \approx u_j$''—such approximations are invalid in formal identification proofs or any mathematical derivation. Even if smoothness is pursued, the key question is how to design a kernel suitable for the latent variable $g$, which demands additional assumptions on its functional form (e.g., a Lipschitz constant).
> > > > >
> > > > > **Response:** We thank the reviewer for this comment. Proposition 1 is intended to answer two narrow and concrete questions that arise after Theorem 1: (i) whether introducing a GP prior in the latent space alters the identifiability of the causal effect (e.g., the ITE), and (ii) whether the invariance of the ITE to invertible reparameterisations $h$ of $U$ is compatible with the GP prior. Proposition 1 shows the answer to both questions: the GP prior leaves identifiability intact, and the latent reparameterisation freedom described in Theorem2 is fully compatible with GP modelling through an induced kernel.
> > > > >
> > > > > Thus, Proposition 1 is not a formal identifiability result and does not make claims about the structural proxy functions $g_i$ or their smoothness. As discussed in our response to the earlier comment on GPs, the smoothness induced by the RBF kernel operates solely on the mapping $Z \mapsto \hat U$ produced by the encoder. It has no bearing on the structural functions $g_i$ assumed in Theorem 1. Its purpose is to regularise the latent inverse problem and provide stable uncertainty propagation, not to characterise the underlying data-generating mechanism.
> > > > >
> > > > > In this context, the ''similar inputs induce similar latent values'' statement in the proof is not part of an identifiability argument but describes the regularisation behaviour of the GP prior in the latent space. The proposition simply formalises that this latent-space coherence does not interfere with the ITE estimation pipeline, and that any invertible transformation of $U$ can be absorbed by an induced kernel $K_h$ without changing the observational or causal content of the model.
> > > > >
> > > > > We agree that the exposition of Proposition 1 and its proof can be made clearer and formally, and in the revision we will restructure it explicitly around the two questions above, removing
> > > > > any wording that could be interpreted as an identifiability claim and emphasising the role of the induced kernel under reparameterisation.
> > > > >
> > > > > Regarding the reviewer's question about kernel design, we emphasise that Proposition 1 does not require constructing a kernel tailored to the structural functions $g_i$. The kernel operates solely in the latent inference space (as discussed in our response above), where it defines a notion of coherence between proxy observations rather than modelling properties of $g_i$ itself. Different kernels (e.g., Matérn, rational quadratic, or non-stationary kernels) can be used interchangeably in this role without altering identifiability or the structure of Proposition 1. The choice of the RBF kernel in our implementation
> > > > > reflects a practical and widely used default for latent regularisation, and we will clarify this design space in the revision.
> > > > >
> > > > > **Comment 6:** LLM Writing Suspicion (for Theorem 1 and Potentially More): The terms "smooth" and "injective" are invoked at least five times before the proof of Theorem 1 (including in the statement itself). This is peculiar, as they bear no relation to the actual proof content. Assuming LLM involvement, this pattern aligns with common motifs in nonlinear ICA literature, a prominent area in machine learning (see below). The paper's overall "plausible but deeply flawed" quality mirrors experiences with LLM-generated text, warranting further scrutiny.
> > > > >
> > > > > **Response:** As clarified in our detailed response above, injectivity at the level of conditional distributions plays an essential role in ensuring the variability required for the Kruskal-rank bounds used in the identifiability argument. These assumptions are therefore justify why the conditional matrices constructed from the proxies contain sufficiently distinct rows to satisfy the conditions of Kruskal's theorem. The original manuscript did not articulate this connection with enough precision, which we acknowledge and will improve in the revision, but the assumptions were not extraneous, nor were they unrelated to the proof structure.We reaffirm our LLM usage disclosure.

---

> > > > > > ### Author Response · Authors · 2025-11-21
> > > > > >
> > > > > > **Comment 7:** "ITE" is used without formal definition; it appears to refer to the "treatment effect" in lines 102–103, but this is not the Individual Treatment Effect (ITE), which should be a deterministic function of all outcome-affecting variables (observed, hidden, and exogenous noise). This is a frequent misnomer in machine learning literature.
> > > > > >
> > > > > > **Response:** We thank the reviewer for raising this point. Since the central objective of our paper is the estimation of Individual Treatment Effects (ITEs), our model must follow the standard counterfactual formulation. According to the literature, a true personalised counterfactual requires an abduction step in which the model recovers the full latent state of each individual that explains the observed outcome. This includes both: (i) the latent confounder, and (ii) the individual-specific exogenous variation entering the outcome equation.
> > > > > >
> > > > > > In our structural equation (1) in the paper,
> > > > > > $$
> > > > > > Y = f(T, U, Z_Y) + \epsilon_Y
> > > > > > $$
> > > > > >
> > > > > > the exogenous noise term $\epsilon_Y$ enters additively. Under Pearl's  abduction–action–prediction semantics, a personalised counterfactual keeps $\epsilon_Y$ fixed when the treatment is changed. Therefore,
> > > > > > $$
> > > > > > y_i(1) = f(1, U_i, Z_{Y,i}) + \epsilon_{Y,i}, \quad
> > > > > > y_i(0) = f(0, U_i, Z_{Y,i}) + \epsilon_{Y,i}
> > > > > > $$
> > > > > > and the Individual Treatment Effect is
> > > > > > $$
> > > > > > \tau_i = y_i(1) - y_i(0)
> > > > > >        = f(1, U_i, Z_{Y,i}) - f(0, U_i, Z_{Y,i})
> > > > > > $$
> > > > > > since the noise terms cancel. This is why $\epsilon_Y$ does not appear explicitly in our definition of ITE.
> > > > > >
> > > > > > The latent variable $U$ in our model represents the confounding structure recoverable from proxy variables, while $Z_Y$ captures outcome-specific latent variation. The additive noise $\epsilon_Y$ remains separate, but does not affect counterfactual contrasts. We can clarify this in the manuscript to avoid ambiguity and to make the connection to the structural ITE explicit.
> > > > > >
> > > > > > **Comment 8:** Typo: In the boxed statement of Theorem 1, the domain of $g_i$ should be $\mathbb{R}$, not $\mathbb{R}^d$.
> > > > > >
> > > > > > **Response:** We thank the reviewer for raising this point. In Theorem 1 we intentionally consider the general case where the latent confounder is $U \in \mathbb{R}^d$. Consequently,
> > > > > > each proxy function is defined as $g_i : \mathbb{R}^d \to \mathbb{R}$. The scalar case ($d=1$), where $g_i : \mathbb{R} \to \mathbb{R}$, arises as a special case of this general formulation.
> > > > > >
> > > > > > However, we acknowledge that the notation in the boxed statement of Theorem 1 was imprecise. The injectivity condition concerns the ``family of conditional distributions'' $\{ p(Z_i \mid U = u) \}_{u \in \mathbb{R}^d}$. Specifically, the intended assumption is that distinct latent values $u \neq u'$ produce distinguishable proxy distributions, i.e. the mapping $u \mapsto p(Z_i \mid U=u)$ is injective in the sense of conditional variability. This corresponds to the“variability” condition.
> > > > > >
> > > > > > We will revise the statement of Theorem 1 to reflect this correctly by removing the misleading functional notation and replacing it with the precise assumption that $p(Z \mid U=u)$ varies injectively in $u$.

---

> > > > > > > ### Author Response · Authors · 2025-11-21
> > > > > > >
> > > > > > > **Comment 9:** All the identifiable or causal VAE papers below are fundamentally based on smooth and injective decoder maps. In particular, the idea of "injective proxy" was considered in *[3]* (Definition 15).
> > > > > > >
> > > > > > > - **[1]** Khemakhem, Ilyes, et al. *Variational Autoencoders and Nonlinear ICA: A Unifying Framework.* AISTATS, PMLR (2020).
> > > > > > >
> > > > > > > - **[2]** Wu, Pengzhou Abel, and Kenji Fukumizu. *β-Intact-VAE: Identifying and Estimating Causal Effects under Limited Overlap.* ICLR (2022).
> > > > > > >
> > > > > > > - **[3]** Wu, Pengzhou, and Kenji Fukumizu. *Towards Principled Causal Effect Estimation by Deep Identifiable Models.* arXiv:2109.15062 (2021).
> > > > > > >
> > > > > > >
> > > > > > > **Response:** We agree with the reviewer that the notion of an "injective proxy" and the associated variability requirements have appeared in several prior contexts, including nonlinear ICA
> > > > > > > and work on prognostic-score learning. These approaches all build on the idea of variability: when an auxiliary or modulation variable induces sufficiently rich variation in the conditional distributions of a latent representation (e.g. $p(Z \mid U)$), that representation can be identified up to an appropriate invertible transformation. Our work also makes use of this general principle, and in the
> > > > > > > revised manuscript we will cite these additional references explicitly. We emphasise, however, that we do not claim the variability condition itself to be novel; rather, it forms a necessary ingredient within our broader identifiability result for proxy-based latent confounder recovery.
> > > > > > >
> > > > > > > Although the variability principle is shared across these literatures, the identification problems they address are conceptually distinct from ours. In nonlinear ICA, for example, the auxiliary variable $u$ is observed, and identifiability relies on how the entire conditional distribution of the latent sources changes as $u$ varies. For each fixed value of $u$, the model generates a full distribution over the latent components, and the crucial requirement is that these distributions vary in sufficiently rich ways across different values of $u$. Methods such as iVAE make explicit use of this observed auxiliary to recover the latent sources.
> > > > > > >
> > > > > > > By contrast, in our setting concerns proxy-based latent confounder recovery, the situation is the *opposite*. The hidden confounder $U$ is unobserved, and must be recovered (up to an invertible transform) from multiple proxy variables $Z_i, i \in \{1, \ldots k\}$, generated from $U$. Our goal is not source separation but the recovery of latent confounders for personalised treatment effect estimation. In this context, the variability assumption is necessary but not sufficient. One must additionally determine how many conditionally independent proxies are required to recover $U$ from its measurements. Our Theorem 1 provides exactly this: a precise proxy-count requirement and an identifiability guarantee for the latent confounder in the hidden-confounding setting. This problem is not addressed by those related papers.

---

### Author Response · Authors · 2025-11-21

We are sincerely grateful to the reviewers for their time and insightful feedback. Their comments have helped identify several aspects of the paper, particularly the presentation of certain theorems and their proofs where the exposition can be substantially improved. We fully acknowledge these issues and will revise the manuscript to strengthen clarity and rigour. At the same time, some concerns appear to arise from misconceptions regarding our theoretical assumptions or the roles of individual model components. We therefore provide clarifications
and detailed point-by-point responses below.

---

> ### Comment · Reviewer_q5kw · 2025-11-22
>
> Just checking that do you plan to update the revision as part of the rebuttal?

---

> > ### Author Response · Authors · 2025-11-25
> >
> > Dear Reviewer, we are currently revising the paper and are carefully addressing your comments. We will update the manuscript before the deadline. Many thanks.

---

> > > ### Author Response · Authors · 2025-12-03
> > > **Submission Withdrawal and Appreciation for Reviewer Feedback**
> > >
> > > Dear Reviewers,
> > >
> > > Due to the issue involving leaked reviewer/AC identities, you will no longer be able to modify your scores or participate in the rebuttal discussion. Given this situation, and considering that the initial scores are not sufficiently high to influence the final decision, we have decided not to submit a revised version as previously planned. Instead, we will withdraw the paper and resubmit it to another appropriate venue.
> > >
> > > We sincerely appreciate your thoughtful feedback, which has substantially improved the quality of our work. Thank you very much for your time and effort.
> > >
> > > Submission 3743 – Authors

---

### Note · Authors · 2025-12-03

I have read and agree with the venue's withdrawal policy on behalf of myself and my co-authors.